# IL-36 promotes anti-viral immunity by boosting sensitivity to IFN-α/β in IRF1 dependent and independent manners

Peng Wang [1], Ana M. Gamero[2] & Liselotte E. Jensen [1]

The functions of the IL-36 cytokines remain poorly understood. We report a previously unrecognized mechanism whereby IL-36 promotes innate antiviral immunity in mouse and human models of herpes simplex virus-1 (HSV-1) infections. HSV-1 actively suppresses production of type I interferon (IFN); our data reveal that IL-36 overcomes this immune evasion strategy by increasing cellular sensitivity to IFN. IL-36β deficient mice display impaired IFN responses and poorly restrict viral replication in skin keratinocytes. In mouse and human keratinocytes IL-36 elicits an antiviral state driven by STAT1 and STAT2 via enhanced expression of IFNAR1 and IFNAR2 subunits of the type I IFN receptor. The degree of IFN regulatory factor 1 (IRF1) involvement is species dependent, with IRF1 playing a more prominent role in human cells. Similar mechanisms are activated by IL-1. Overall, IL-36 acts as an antiviral cytokine by potentiating type I IFN signaling and thereby upholds immune responses to viruses that limit the production of IFNs.

---

[1] Department of Microbiology and Immunology, Temple University Lewis Katz School of Medicine, Philadelphia, PA 19140, USA. [2] Department of Medical Genetics and Molecular Biochemistry, Temple University Lewis Katz School of Medicine, Philadelphia, PA 19140, USA. Correspondence and requests for materials should be addressed to L.E.J. (email: liselott@temple.edu)

Cytokines are essential for orchestrating both innate and adaptive immune responses against microorganisms. The interleukin-36 cytokines (IL-36α, IL-36β, and IL-36γ) are related to IL-1α and IL-1β and are part of the IL-1 superfamily, which also includes IL-18 and IL-33[1]. The IL-36s have been implicated in immunity against viruses; however, the mechanisms behind their both protective and detrimental effects remain to be elucidated.

All three IL-36s utilize the heterodimeric IL-36 receptor, comprising IL-1 receptor like-2 (IL-1RL2) and IL-1 receptor accessory protein (IL-1RAP), for activation of intracellular signaling pathways and gene transcription. The IL-1s also signal via IL-1RAP, but use IL-1R1 as their binding receptor (reviewed in Jensen et al.[2]). While the IL-1 system, comprising the cytokines and their receptor, is highly abundant, the IL-36 system is more restricted with an expression profile that suggests functions of most impact to epithelial tissues[3]. Epithelial cells are common targets of viral infections, and we previously proposed that the IL-1 superfamily arose during evolution to counteract microbial immune evasion mechanisms that limit IL-1 and IL-18 functions[2].

Influenza viruses restrain activation of IL-1β through NS1 mediated inhibition of the inflammasome and caspase-1[4–6]. Studies of influenza lung infections in mice have provided somewhat conflicting data regarding the involvement of the IL-36s in infection progression and disease pathogenesis. One study showed that IL-36 contributes to lung damage and mortality by promoting inflammation;[7] another reported that IL-36 protects against mortality and limits viral replication, at least in part, by facilitating survival of macrophages in the lungs[8].

The herpes simplex viruses, HSV-1 and HSV-2, are common and establish life-long infections of neurons. During latency HSV exhibits minimal viral activity in infected neurons to avoid detection by the host. However, active disease can be triggered by a number of factors leading to viral replication and spread to epithelial tissues, typically the oral and genital mucosa, where lesions form. At these sites, HSV replication is generally well-controlled and the virus is cleared within a few weeks[9]. However, in children with eczema or patients with immune deficiencies, these viruses can be detrimental as they spread to vital organs such as the brain, liver and lungs leading to permanent tissue damage and even death[10,11].

Like influenza, HSV has acquired immune evasion strategies to limit activation of IL-1β. The best-defined mechanism involves HSV ICP0 mediated degradation of the DNA sensor IFI16, which is essential for activation of the inflammasome[12,13]. Furthermore, HSV infected cells appear to retain IL-1β through a poorly understood mechanism[13,14]. Interferons (IFNs) are another type of cytokines essential for innate antiviral mechanisms as they stimulate production of proteins that directly restrict viral replication[15]. Viruses, including HSV, have developed several schemes to inhibit the production and function of these IFNs (reviewed in refs. [9,16]).

The IL-36 cytokines were previously shown to have protective activity against HSV-1 skin infections[17] and genital HSV-2 infections[18,19]. Nevertheless, the specific mechanism behind this activity has not been elucidated. We here describe an IL-36 regulated innate immune pathway that enables keratinocytes to respond more efficiently to low levels of type I IFN. Through this mechanism, IL-36 cytokines provide accessory help to overcoming viral immune evasion mechanisms that inhibit IFN production; this ensures rapid and effective immune responses.

## Results

**IL-36β restricts HSV-1 replication in mouse skin.** Our previous studies of viral skin disease revealed that IL-36 knockout (KO) mice develop larger zosteriform lesions along affected dermatomes following flank skin HSV-1 infection[17]. Such larger lesions could be explained by delayed immune responses or wound healing. The former would directly affect early levels of the virus, while the latter would not. In the flank model, the virus is first introduced through abrasion into the skin, were the virus infects both keratinocytes and sensory nerves. Through retrograde migration the virus: (1) enters the dorsal root ganglia, (2) replicates, (3) spreads anterograde back into the skin, and (4) causes skin lesions at numerous nerve endings along the affected dermatome (illustrated in Milora et al.[17]). In our hands, these lesions first appear on day 5 and continue to grow in size through day 7 in wild type mice[17,20]. To examine if viral loads were different in HSV-1 infected wild type and IL-36β KO mice, skin surrounding the primary infection site and all new lesions (totalling 4 cm$^2$) were collected at day 6 post-infection. In both female and male mice higher levels of HSV-1 DNA were detected in IL-36β KO mice compared to wild type (Fig. 1a). Similar observations were made when the HSV-1 protein ICP4 was examined (Fig. 1b, c). These results suggest that IL-36β plays an important role in restricting HSV-1 replication in the skin.

**IL-36 promotes expression of antiviral ISGs.** To gain insight into the immunological mechanism underlying the observed difference in viral load (Fig. 1a–c), we initially employed a PCR array screening approach. A total of 86 genes involved in antiviral immunity were examined using skin RNA isolated at day 6 post-infection (Supplementary Table 1). Of these genes, 14 were expressed at lower levels in skin from IL-36β KO mice than wild type and one was present at elevated levels (Fig. 1d and Supplementary Table 1). The differentially expressed genes are involved in signaling via the type I IFN, Toll-, NOD- and/or RIG-I-like receptors (Fig. 1e). The type I IFN response is critically involved in regulating expression of proteins with antiviral activity, e.g., Mx1. Since our data suggest a role for IL-36β in restricting viral replication in the skin (Fig. 1a–c), we examined expression of other IFN stimulated genes (ISGs, Fig. 1f). In agreement with the differential expression of Mx1 (Fig. 1d, e), Oas1, Eif2ak2 (also known as Pkr), Isg15, Ifitm2, Ifitm3 and Ifit3 mRNA levels were all lower in IL-36β KO mice when compared to wild type mice (Fig. 1f). We did not identify any noticeable differences between female and male mice (Fig. 1f). In summary, these expression analyses reveal IL-36β dependent regulation of several innate signaling pathways during HSV-1 skin infections.

**STAT activation is reduced in HSV-1 infected IL-36β KO skin.** Our expression analyses of antiviral genes in HSV-1 infected skin indicated impairment in type I IFN signaling in IL-36β KO mice (Fig. 1). STAT1 and STAT2 play important roles in type I IFN signaling and induction of ISGs; hence, we examined levels of STAT1 and STAT2 expression and activation in wild type and IL-36β KO mice following HSV-1 skin infection (Fig. 2). No differences in Stat1 (Fig. 2a) or Stat2 (Fig. 2b) mRNA levels were detected between the two strains of mice. In agreement with this the total levels of STAT1 (Fig. 2c) and STAT2 (Fig. 2d) proteins were also similar. Type I IFN promotes activation of STAT1 and STAT2 through phosphorylation. Our analyses of STAT1 and STAT2 phosphorylation revealed lower levels of activated pSTAT1 (Fig. 2c) and pSTAT2 (Fig. 2d) in IL-36β KO mice than wild type. This suggests that IL-36β plays an important role in promoting STAT1/2 activation during viral skin infections.

**IL-36β enhances STAT1/2 activation in keratinocytes.** The type I IFN response is an innate immune pathway present in most cell types, including leukocytes, fibroblasts, and keratinocytes in the

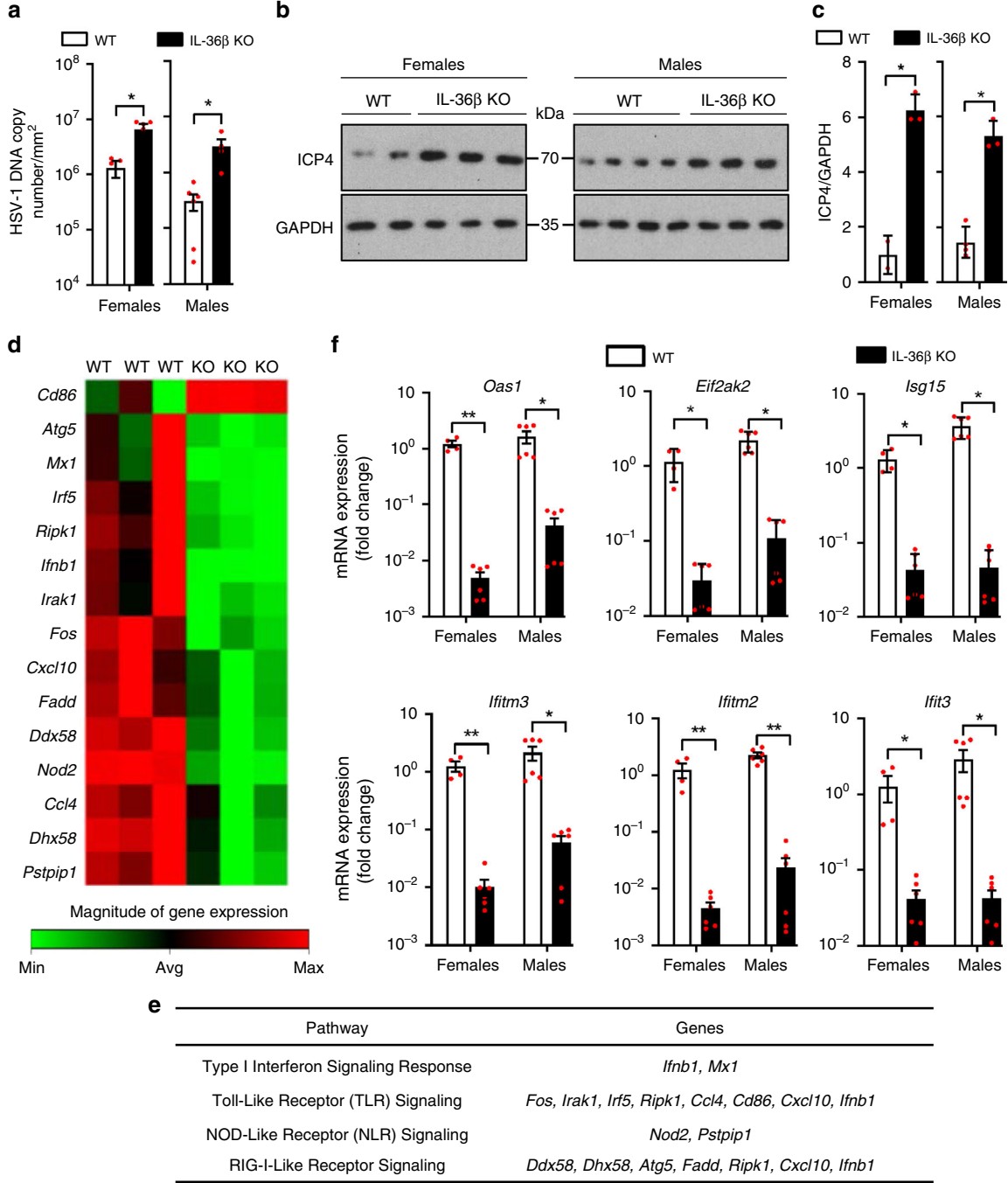

**Fig. 1** IL-36β KO mice exhibit altered responses to HSV-1 skin infection. **a** Wild type (WT; $n = 5$ ♀ and 13 ♂) and IL-36β KO mice ($n = 7$ ♀ and 12 ♂) were infected with HSV-1 on the flank and skin regions along HSV-1 infected dermatomes collected 6 days post-infection. Viral DNA loads were determined by QPCR. Data are pooled from three independent experiments and shown as means ( ± SD). *$p < 0.05$ (One-Way ANOVA). **b, c** Viral ICP4 protein levels were examined by Western blotting (**b**) and quantified using ImageJ software (**c**). GAPDH was used as loading control. Data are representative of three independent experiments and shown as means (±SD) in **c** (WT: $n = 2$ female and 4 male; IL-36β KO mice: $n = 3$ female and 3 male). *$p < 0.05$ (one-way ANOVA). **d** Wild type and IL-36β KO mice were infected with HSV-1 ($n = 3$ per group) on the flank and skin RNA isolated 6 days post-infection. Heat-map of genes differentially expressed in the two strains is shown. **e** Pathway associations of genes identified as differentially expressed in **d**. **f** Expression of *Oas1, Eif2ak2, Isg15, Ifitm3, Ifitm2,* and *Ifit3* mRNAs were examined in wild type and IL-36β KO HSV-1 infected skin (**a**). The mRNAs were normalized against GAPDH and are shown as relative expression compared to female wild-type mice (means ± SD). *$p < 0.05$; **$p < 0.005$ (one-way ANOVA). Each red dot represents a single data point. Source data are provided as a Source Data file

skin. In vivo, it has been shown that the basement membrane between the epidermis and dermis prevents spread of HSV-1 from the former to the latter[21]. To determine which cells are activated in our model, we examined localization of pSTAT1 in uninfected and HSV-1 infected mouse skin using immunohistochemistry (Supplementary Fig. 1). Nuclear pSTAT1 staining was observed distinctively in keratinocytes in the epidermis and hair follicles proximal to HSV-1 lesions (Supplementary Fig. 1i, j). Some immune cells under the wound scabs, in the dermis and subcutaneous fat were also positive

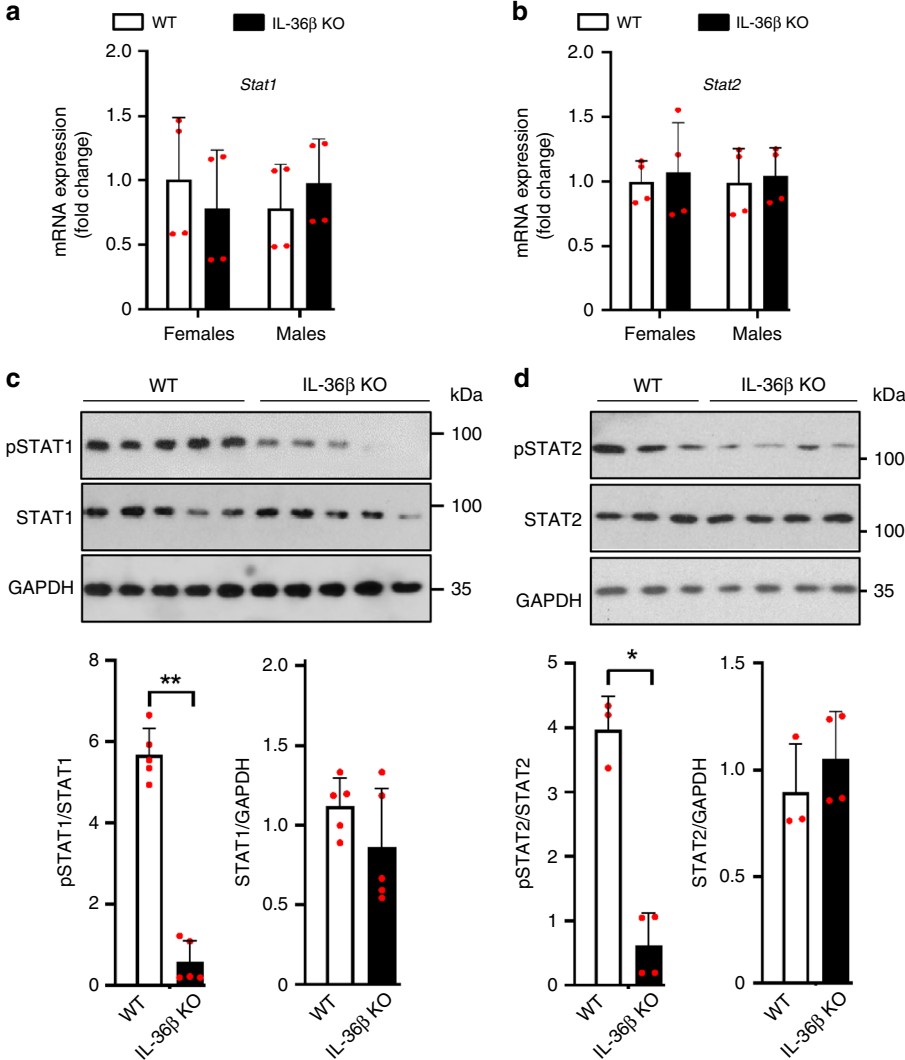

**Fig. 2** IL-36β promotes activation of STAT1 and STAT2 during HSV-1 skin infection. **a**, **b** Expression of *Stat1* (**a**) and *Stat2* (**b**) mRNAs were examined in wild type and IL-36β KO HSV-1 infected skin (Fig. 1a). No statistically significant differences were detected. **c** Quantification of total STAT1 and pSTAT1 in wild type and IL-36β KO HSV-1 infected skin by western blotting and ImageJ analysis (WT, $n = 5$; KO, $n = 5$). **d** STAT2 and pSTAT2 levels in wild type and IL-36β KO HSV-1 infected skin were determined by western blotting and ImageJ analysis (WT, $n = 3$; KO, $n = 4$). **c**, **d** Representative data from one of three independent experiments involving male mice is shown. Quantitative data are shown as means ± SD. *$p < 0.05$; **$p < 0.01$ (one-way ANOVA). Each red dot represents a single data point. Source data are provided as a Source Data file

(Supplementary Fig. 1k, l). Uninfected skin showed no pSTAT1 staining (Supplementary Fig. 1g).

Next STAT1 activation in wild type and IL-36β KO mice was compared. Here we focused on the zosteriform infections as: (1) pure HSV-1 infections without mechanical wounding required for initial infection in the model, and (2) the sites in which we found differences in viral titers (Fig. 1a, b). Positive pSTAT1-cells were present within epidermal regions of keratinocytes exhibiting early signs of cytopathic effects of the HSV-1 infection in both strains (Fig. 3a–h). STAT1 was also activated in keratinocytes in the epidermis proximal to intermediate lesions where the cytopathic effects were more advanced, yet the epidermis still remained intact (Fig. 3i–p). These regions grossly appeared to have more nuclear pSTAT1 in wild type (Fig. 3k, l) than IL-36β KO mice (Fig. 3o, p). This is in agreement with our observations using the more quantitative Western blotting approach (Fig. 2c); however, further analyses will be required to determine if these intermediate lesion proximal sites are indeed responsible for overall greater STAT1

activation in wild type than IL-36β KO mice. In more advanced lesions, where the epidermis had been destroyed, we also found pSTAT1 positive keratinocytes in the surrounding epidermis; however, this staining was too scarce to evaluate potential differences between the mouse strains. Additional STAT1 activation was seen in some immune cells recruited into the subcutaneous fat and dermis (Fig. 3, red arrows); however, the immunohistochemistry approach did not allow us to discern the degree of this activation.

As HSV-1 replicates in epithelial cells, we focused our further mechanistic studies on keratinocytes. We first examined whether IL-36β could modulate the degree of STAT1 and STAT2 activation in human keratinocytes (Supplementary Fig. 2) resembling that seen in vivo in whole mouse skin (Fig. 2). Levels of both pSTAT1 and pSTAT2 were dramatically increased in response to HSV-1 without significant changes in total levels of the individual STAT proteins (Supplementary Fig. 2). Pre-treatment of cells with IL-36β resulted in even higher levels of pSTAT1 and pSTAT2 (Supplementary Fig. 2), indicating that

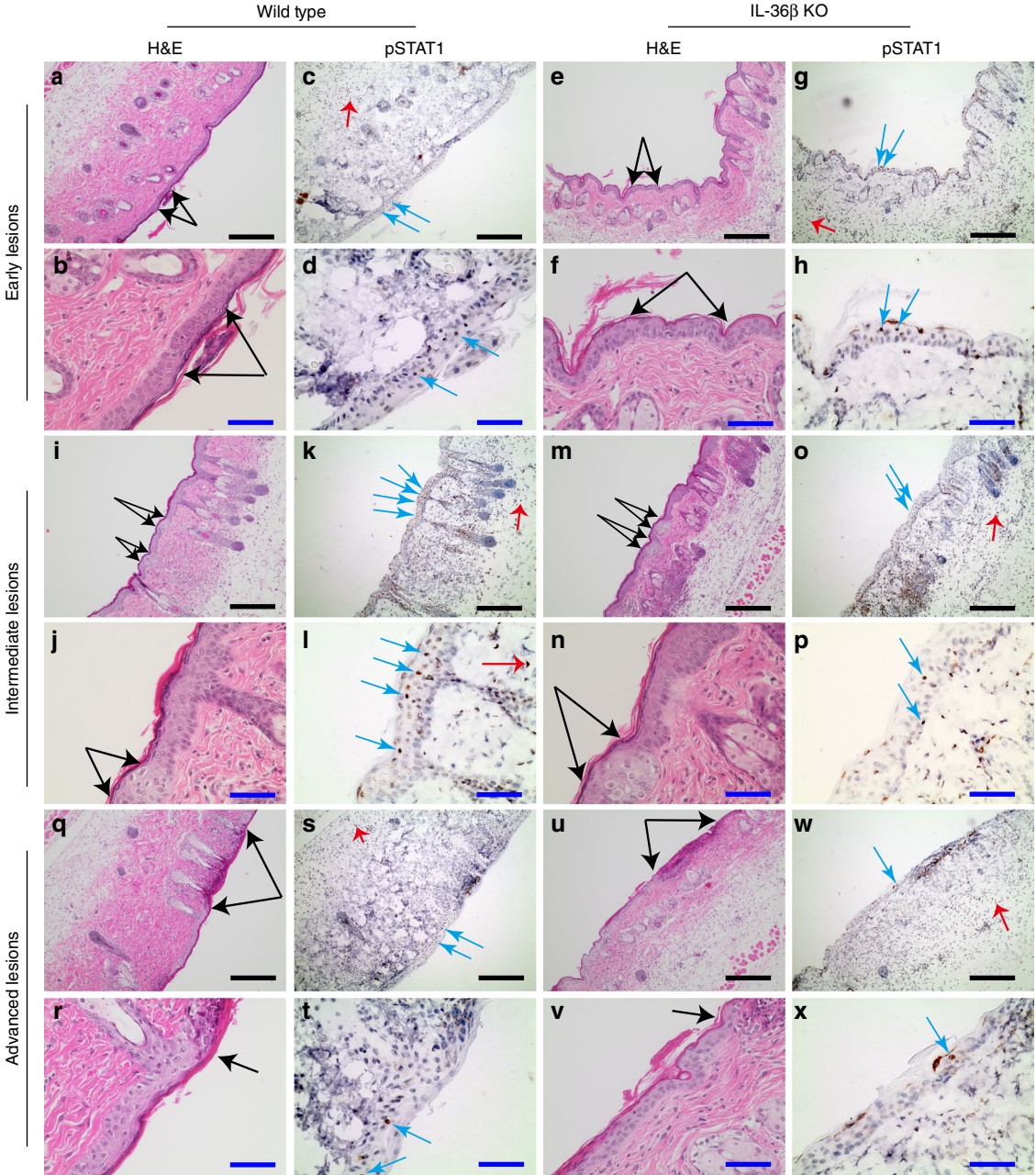

**Fig. 3** STAT1 is activated in epidermal keratinocytes during HSV-1 infection. Wild type and IL-36β KO mice were infected with HSV-1 and skin along the dermatome collected 5 days later. Consecutive skin sections were examined by H&E and pSTAT1 immunohistochemistry. Early (**a–h**), intermediate (**i–p**), and advanced (**q–x**) lesions are shown. Black, blue, and red arrows point to lesion edges and positive pSTAT1 nuclei in keratinocytes and leukocytes, respectively. Black and blue scale bars represent 200 and 50 μm, respectively

IL-36β enhances STAT1/2 activation during HSV-1 infection of keratinocytes.

**IL-36β promotes viral resistance in keratinocytes**. STAT signaling is essential for many innate antiviral mechanisms[15]. Since IL-36β enhanced STAT1 and STAT2 activation during HSV-1 infection (Supplementary Fig. 2), we next examined whether exogenously administered IL-36β could also promote an antiviral state. Using both human and mouse keratinocytes, we observed diminished HSV-1 levels in cells that were pre-treated with IL-36β (Fig. 4a, b), suggesting that IL-36β limits the ability of the virus to replicate. We next examined whether endogenous IL-36β could promote a similar response. Comparison of primary wild type and IL-36β deficient mouse keratinocytes revealed higher

levels of HSV-1 ICP4 protein in IL-36β KO cells than wild type (Fig. 4c). This effect appeared similar in female and male keratinocytes (Fig. 4c). Overall, our data suggest that IL-36β promotes antiviral mechanisms in human and mouse keratinocytes.

**IL-36β antiviral activity requires STAT1 and STAT2**. Given the observations that IL-36β enhances STAT1/2 activation during HSV-1 infection of keratinocytes (Supplementary Fig. 2) and promotes viral resistance (Fig. 4a–c), we next tested whether the latter was dependent upon the former. Since our in vivo studies identified Mx1 as an IL-36β regulated gene during HSV-1 infection (Fig. 1), we employed this gene as a general marker of ISG activation. As expected, using primary keratinocytes derived from wild type and STAT1 KO mice, increased replication of HSV-1 was observed in

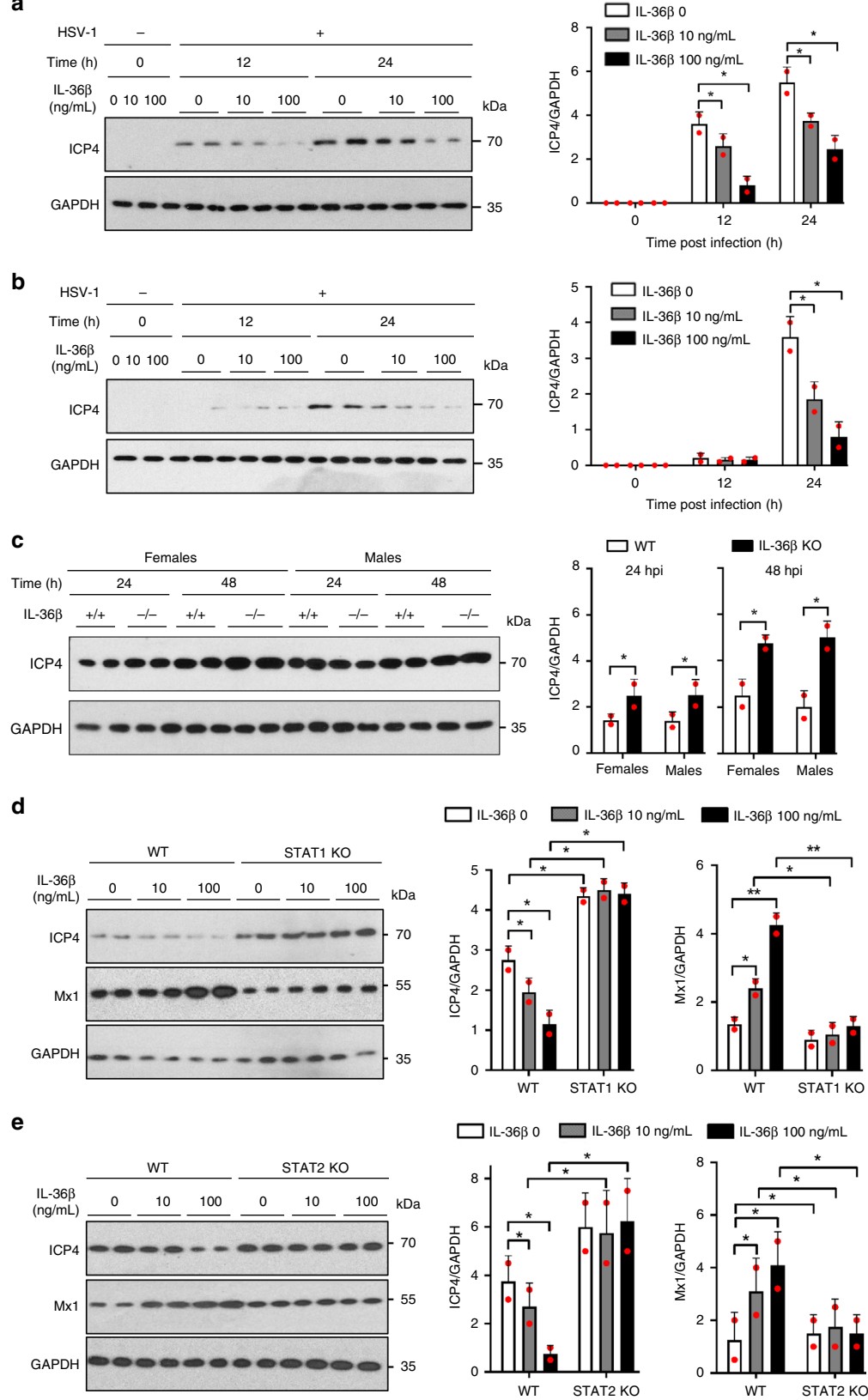

the STAT1 deficient male (Fig. 4d) and female cells (Supplementary Fig. 3a). A similar trend was observed using STAT2 KO male (Fig. 4e) and female cells (Supplementary Fig. 3b). Furthermore, in both the STAT1$^{-/-}$ and STAT2$^{-/-}$ keratinocytes, IL-36β no longer induced an antiviral state (Fig. 4d, e and Supplementary Fig. 3).

Correlating with these observations, expression of Mx1 was not enhanced by IL-36β in neither STAT1$^{-/-}$ or STAT2$^{-/-}$ cells (Fig. 4d, e and Supplementary Fig. 3). Thus, ISG expression and antiviral mechanisms induced by IL-36β are dependent upon both STAT1 and STAT2.

**Fig. 4** IL-36β induces STAT1- and STAT2-dependent antiviral immunity in keratinocytes. **a** Human keratinocytes were pre-treated with medium only or IL-36β before infection with HSV-1 (MOI = 0.01). Levels of HSV-1 ICP4 protein were determined by Western blotting and ImageJ analyses using GAPDH as control. **b** Mouse primary keratinocytes were pre-treated with medium only or IL-36β, followed by HSV-1 infection (MOI = 0.01), and ICP4 levels examined by western blotting. **c** Wild type ( + / + ) and IL-36β KO (−/−) mouse primary keratinocytes were infected with 0.01 MOI HSV-1 and ICP4 examined by western blotting. **d** Wild type and STAT1$^{-/-}$ primary male mouse keratinocytes were treated with medium only or IL-36β followed by HSV-1 infection (MOI = 0.01) for 24 h. Levels of HSV-1 ICP4 and host Mx1 were examined by western blotting. **e** Wild type and STAT2$^{-/-}$ primary male mouse keratinocytes were examined after IL-36β pre-treatment and HSV-1 infection using western blotting. **a–e** Quantitative data are shown as means ± SD. *$p < 0.05$ (one-way ANOVA, $n = 2$ biologically independent samples per treatment group); **$p < 0.01$. Each red dot represents a single data point. Source data are provided as a Source Data file

## IL-36β induces IFNAR expression in keratinocytes.
Type I IFNs induce STAT1 and STAT2 phosphorylation by engaging their heterodimeric receptor comprising IFN-α and -β receptor 1 (IFNAR1) and IFNAR2[15]. Hence, we next examined if IL-36β could regulate expression of the IFNAR genes and proteins. Using mouse primary keratinocytes, we found both the *Ifnar1* and *Ifnar2* mRNAs to be upregulated by IL-36β in concentration and time-dependent manners (Fig. 5a and Supplementary Fig. 4a). Upregulation of the IFNAR proteins followed a similar pattern (Fig. 5b and Supplementary Fig. 4b). Comparable observations were made using human keratinocytes (Fig. 5c, d). Thus, our data demonstrate that IL-36β is capable of increasing expression of the type I IFN receptor in both human and mouse cells.

## IRF1 contributes differentially to IFNAR expression.
Previously, microarray studies identified *IRF1* as an IL-1 induced gene in keratinocytes[22] and in silico analyses proposed IRF1 as a regulator of *IFNAR2*[23]. To examine the potential role of IRF1 in regulating expression of IFNAR, we first tested whether IL-36β could induce expression of IRF1. Expression of IRF1 mRNA was indeed found to be upregulated by IL-36β in both mouse (Fig. 6a) and human (Fig. 6b) keratinocytes. We next evaluated IFNAR expression in mouse wild type and IRF1 KO cells (Fig. 6c, d). Interestingly, while expression of the *Ifnar2* mRNA was not affected by the absence of IRF1, levels of *Ifnar1* mRNA were significantly lower in IRF1 KO cells than wild type after IL-36β treatment (Fig. 6c and Supplementary Fig. 5). Despite the reduction in *Ifnar1* mRNA expression, the mRNA was still induced by IL-36β in IRF1 KO cells (Fig. 6c). Similar analyses of IFNAR protein levels did not reveal significant differences (Fig. 6d). Hence, while *Ifnar2* expression is independent of IRF1, IRF1 plays a minor role in increasing gene expression of *Ifnar1* mRNA in response to IL-36β in mouse keratinocytes. However, this latter effect may not affect functional outcome as protein levels are not impacted by the absence of IRF1.

The absence of IRF1 involvement in mouse *Ifnar2* mRNA expression was surprising given the earlier reported in silico studies of human cells[23]. Therefore, we next performed similar experiments in human keratinocytes in which *IRF1* expression was deleted through CRISPR/Cas9 gene editing (Fig. 6e, f and Supplementary Fig. 6). Interestingly, in the human cells IL-36β induction of both *IFNAR1* and *IFNAR2* was significantly reduced in the IRF1$^{-/-}$ cells; however, both genes were still activated in response to IL-36β (Fig. 6e and Supplementary Fig. 6a). Similar observations were made when IFNAR protein levels were analyzed (Fig. 6f and Supplementary Fig. 6b). Thus, human IFNAR1 and IFNAR2 are both regulated by IL-36β through IRF1 dependent and independent mechanisms. In contrast, IRF1 does not impact mouse IFNAR1 and IFNAR2 protein levels.

## The role of IRF1 in antiviral immunity is species dependent.
In light of our data demonstrating that IL-36β induced mouse *Ifnar1* mRNA is partially dependent upon IRF1 in keratinocytes (Fig. 6c), we further examined if this effect would be sufficient to impact immunity in vivo in mice. In our HSV-1 flank model, the mean survival time was >16 days for wild type mice, but only 12 days in the IRF1 KO mice (Fig. 7a). The greater mortality in the IRF1$^{-/-}$ mice was associated with a trend towards earlier weight loss (Fig. 7b). Previously, we observed severe bowel dysfunction involving especially the small intestine in moribund HSV-1 flank-infected wild type, IL-1R1$^{-/-}$ and IL-36β$^{-/-}$ mice[20]. Others reported constipation in mice with vaginal HSV-2[24] and HSV-1[25] infections and the latter was linked to mortality. Here we found that at day 9 post-HSV-1 flank infection, 5 of 6 examined IRF1 KO mice had gastrointestinal dysfunction (Fig. 7c); in some mice this even involved the stomach (Fig. 7c, red arrow). In contrast, only 3 out of 14 wild type mice had signs of disease in the small intestine (Fig. 7c). No signs of paralysis were observed.

Skin lesions were also larger in the IRF1 KO mice than in wild type male mice (Fig. 7d) with a similar trend observed in female mice (Supplementary Fig. 7a; note that only a small number of female mice were available for analyses). Interestingly, HSV-1 DNA copy numbers were not significantly different in neither male nor female mice (Fig. 7e and Supplementary Fig. 7b). Since HSV-1 primarily infects keratinocytes in the skin, our data could suggest that IRF1 is not essential for innate immunity within keratinocytes, and we therefore next examined HSV-1 infection of primary keratinocytes derived from wild type and IRF1 KO mice (Fig. 7f). While infection of wild type and IRF1$^{-/-}$ keratinocytes resulted in higher ICP4 protein levels in the IRF1$^{-/-}$ cells in one experiment (Fig. 7f), this was not reproducible in other experiments. In contrast, IL-36β induced resistance to HSV-1 infection, as determined through ICP4 levels, was repeatedly present in both cell populations (Fig. 7f). Interestingly, IL-36β did not induce more Mx1 in IRF1 KO cells (Fig. 7f). Hence, our data from the mouse cells suggest that while IRF1 is important for IL-36β enhanced Mx1 expression, IRF1 is dispensable for the antiviral state induced by IL-36β. This is likely due to IL-36β inducing multiple antiviral proteins (Fig. 1f) that compensate for Mx1.

In extension of our surprising data revealing differential regulation of IFNAR1 and IFNAR2 expression in mouse and human cells (Fig. 6), we further examined antiviral immunity in CRISPR/Cas9 control and *IRF1* targeted human keratinocytes. Unlike the mouse cells (Fig. 7f), the IL-36β induced antiviral state was absent in the human IRF1$^{-/-}$ cells (Fig. 7g), i.e. human keratinocytes are dependent upon IRF1 for IL-36β enhanced innate immunity. Furthermore, Mx1 expression was not increased by IL-36β when IRF1 was absent (Fig. 7g).

Overall, our observations demonstrate that while IRF1 plays a critical role in determining disease outcome (Fig. 7a–d), IRF1 is not essential for innate antiviral immunity in the skin of the mouse (Fig. 7e–f). By contrast, human keratinocytes are dependent upon IRF1 for induction of a productive innate immune response within these cells (Fig. 7g).

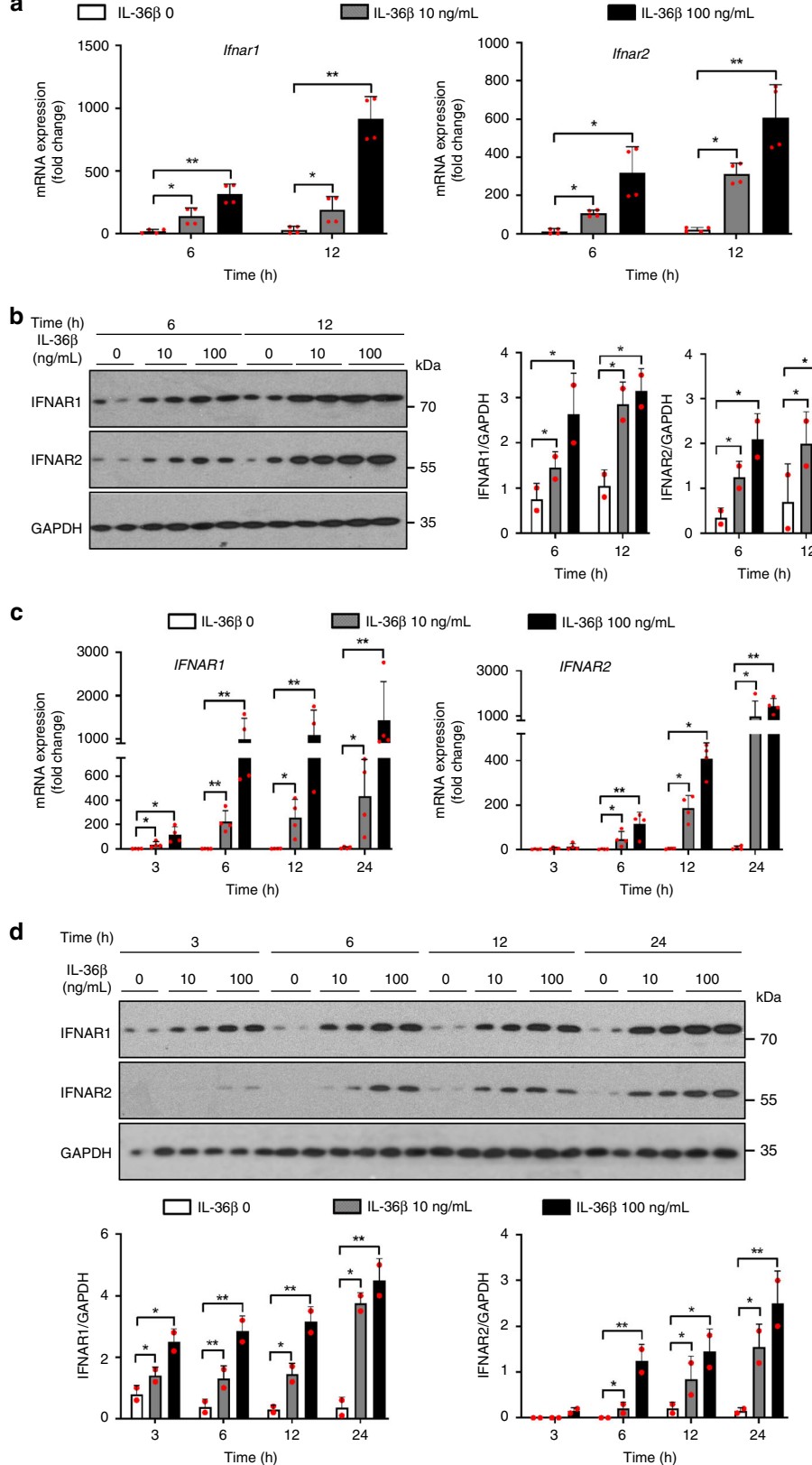

**Fig. 5** IL-36β activates expression of IFNAR1 and IFNAR2. **a** *Ifnar1* and *Ifnar2* mRNA expression was analyzed by real-time PCR in male mouse primary keratinocytes treated with medium only or IL-36β as indicated. **b** Mouse IFNAR1 and IFNAR2 protein expression was examined by western blotting and ImageJ analyses. **c** Human keratinocytes were treated with medium only or IL-36β and expression of *IFNAR1* and *IFNAR2* mRNA determined by real-time PCR. **d** Expression of human IFNAR1 and IFNAR2 protein was examined by western blotting and ImageJ analyses. **a–d** Quantitative data are shown as means ± SD. *$p < 0.05$ (one-way ANOVA, $n = 2$ biologically independent samples per treatment group); **$p < 0.01$. Each red dot represents a single data point. Source data are provided as a Source Data file

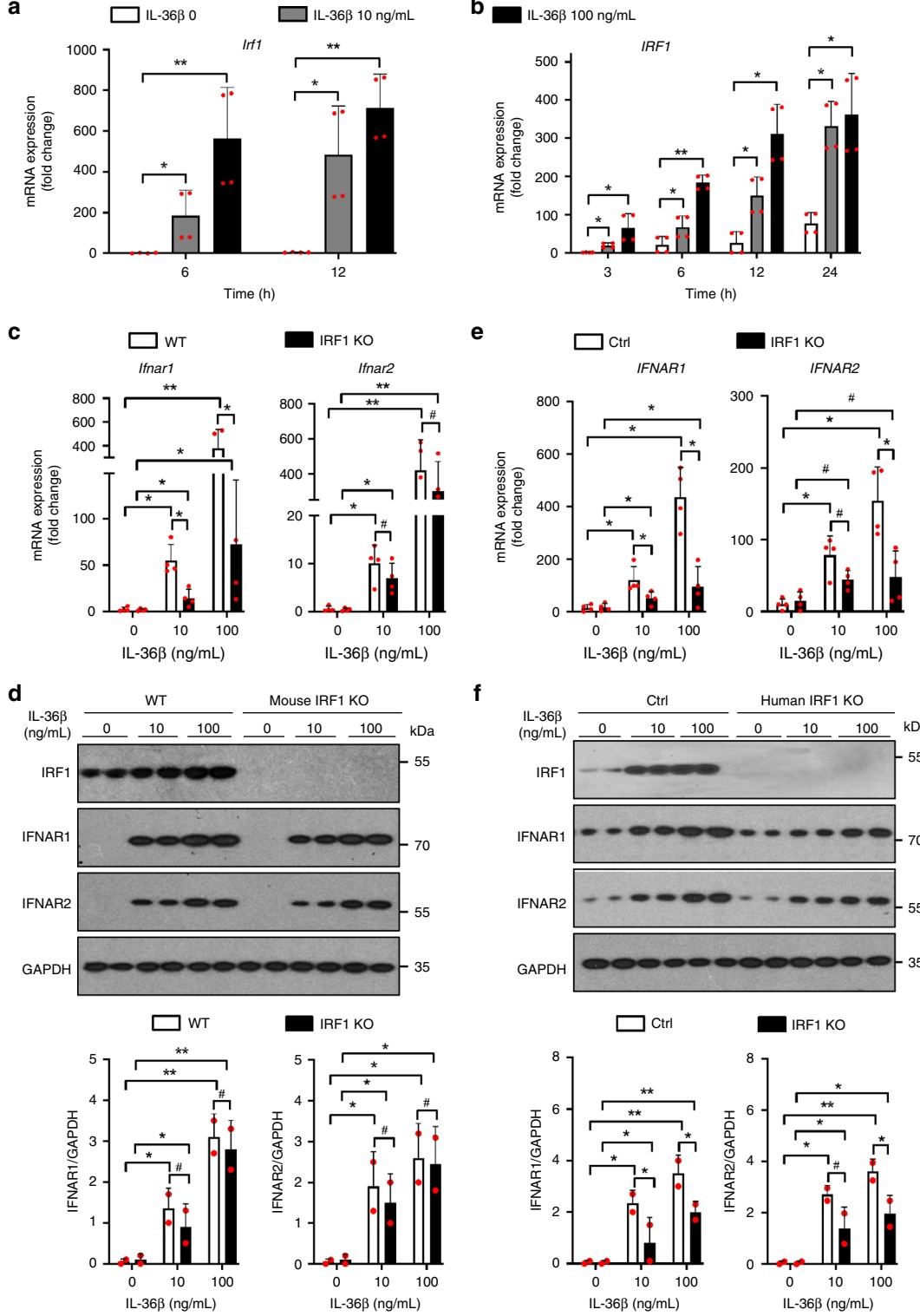

**Fig. 6** IRF1 is induced by IL-36β and has divergent impact on IFNAR expression. **a**, **b** *IRF1* mRNA expression in mouse (**a**) and human (**b**) keratinocytes was examined following IL-36β treatment at indicated time-points. **c** Mouse wild type and IRF1$^{-/-}$ primary keratinocytes were treated as indicated for 6 h and expression of *Ifnar1* and *Ifnar2* mRNA analyzed by real-time PCR. **d** Mouse wild type and IRF1$^{-/-}$ keratinocytes were treated with medium only or IL-36β for 6 h and IFNAR protein expression examined by western blotting and ImageJ analyses. **e** *IFNAR1* and *IFNAR2* mRNA expression in human control (Ctrl) and IRF1$^{-/-}$ keratinocytes was examined following medium only or IL-36β treatment for 6 h. **f** Protein levels of IFNAR1 and IFNAR2 in human control and IRF1$^{-/-}$ keratinocytes were determined by western blotting 6 h post-treatment. **a–f** Quantitative data are shown as means ± SD. *$p < 0.05$; **$p < 0.01$; #$p > 0.05$ (one-way ANOVA, $n = 2$ biologically independent samples per treatment group). Each red dot represents a single data point. Source data are provided as a Source Data file

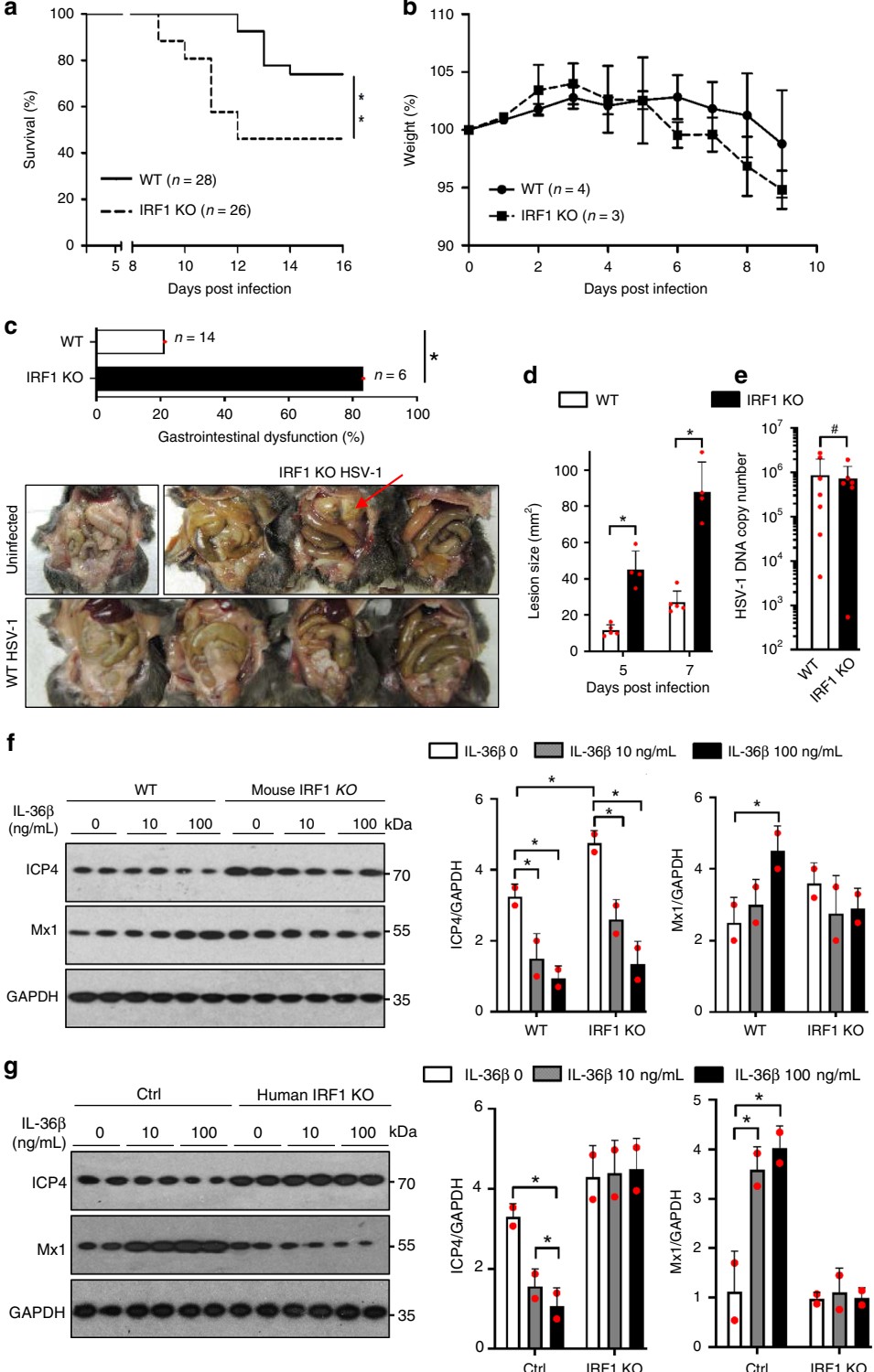

**Fig. 7** IRF1 is partially involved in immunity against HSV-1. **a–e** C57BL/6 J (WT) and IRF1 KO mice were infected with HSV-1 on the flank. **a** Survival was monitored for 16 days. **b** Weight was measured for 9 days. **c** Gastrointestinal dysfunction was examined at day 9 post-infection. Red arrow points to disease affected stomach. **d** Skin lesion sizes were measured (male mice; WT: $n = 5$; KO: $n = 4$). **e** HSV-1 DNA copy numbers in the skin were determined 6 days post-infection (male mice; WT: $n = 7$; KO: $n = 6$). **f** Mouse primary keratinocytes from wild type (WT) and IRF1 KO mice were sequentially treated with medium only or IL-36β as indicated and infected with HSV-1 ($n = 2$ biologically independent samples per treatment group). Levels of ICP4 and Mx1 were evaluated by western blotting and ImageJ analyses. **g** Levels of ICP4 and Mx1 protein were evaluated by western blotting and ImageJ analyses following IL-36β treatment and HSV-1 infection of human control (Ctrl) and IRF1 KO keratinocytes ($n = 2$ biologically independent samples per treatment group). **a**, **c** *$p < 0.05$; **$p < 0.01$ (Mantel-Cox and Gehan-Breslow-Wilcoxon tests). **d–g** Quantitative data are shown as means ± SD. *$p < 0.05$; #$p > 0.05$ (one-way ANOVA). Each red dot represents a single data point. Source data are provided as a Source Data file

**IL-36β promotes antiviral activity through IFNAR**. Given our observations that IL-36β enhances antiviral immunity in keratinocytes in a STAT1/2 dependent manner (Fig. 4d, e) and it induces IFNAR expression (Fig. 5 and Supplementary Fig. 4), we further tested whether the antiviral activity promoted by IL-36β was dependent upon signaling through IFNAR. In human keratinocytes, in which expression of IFNAR1 or IFNAR2 was eliminated using CRISPR-Cas9 gene editing, higher HSV-1 replication was observed when cells were not treated with IL-36β (Fig. 8a, b). Importantly, in IFNAR1 deficient cells IL-36β no longer protected against infection and IL-36β did not induce Mx1 expression either (Fig. 8a). Similar observations were made when IFNAR2 was deleted (Fig. 8b). Analyses using mouse primary keratinocytes and neutralizing antibodies directed against the mouse IFNARs revealed similar outcomes (Fig. 8c). Hence, the antiviral state induced by IL-36β is dependent upon both IFNAR1 and IFNAR2 in both human and mouse cells.

**IL-36β accelerates STAT activation in response to IFN-α/β**. Since IL-36β enhances IFNAR expression (Fig. 5 and Supplementary Fig. 4) and IFNAR binding of type I IFN leads to STAT1 and STAT2 activation essential for IFN induced antiviral activity, we next examine if IL-36β could promote quicker activation of the STATs in response to type I IFN in human keratinocytes (Fig. 9a and Supplementary Fig. 8). Initial preliminary analyses using high concentrations of IFN-α/β that may engage all IFNAR heterodimers revealed trends towards enhanced STAT1/2 activation in response to IL-36β (Supplementary Fig. 8, ≥5 min). Subsequent experiments involving a lower IFN-α/β concentration demonstrated earlier phosphorylation of STAT1 and STAT2 in IL-36β concentration dependent manners (Fig. 9a), i.e. IL-36β accelerated STAT1 and STAT2 activation in response to IFN-α/β. Similar observations were made in mouse primary keratinocytes (Fig. 9b). These observations are in agreements with our in vivo data demonstrating reduced type I IFN signaling in IL-36β KO mice (Fig. 1) and our in vitro data showing elevated STAT1/2 activation in the presence of IL-36β (Supplementary Fig. 2). Note that the here used concentrations of IFN-α/β are within the range of that secreted by HSV-1 infected skin (Supplementary Fig. 9). In summary, our data reveal that IL-36β plays an important role in innate immunity by enhancing sensitivity of epithelial cells towards type I IFN.

**IL-1 also enhances IFN-α/β signaling via IFNAR upregulation**. While the IL-36 and IL-1 cytokines utilize separate receptors for ligand binding, IL-1RL2 and IL-1R1 respectively, they share signaling pathways through IL-1RAP, MyD88, and IRAK[2]. This suggests they regulate the same genes and therefore we next examined if IL-1 can engage the same anti-viral mechanism as that induced by IL-36. Since we previously found that IL-1α is released from HSV-1 infected keratinocytes[20], we focused on this IL-1 isoform. As expected IL-1, like IL-36, stimulated increased expression of IFNAR mRNA and protein in human (Supplementary Fig. 10) and mouse keratinocytes (Supplementary Fig. 11). In agreement with this and our IL-36 related observations described above, IL-1 promoted an antiviral state that was IFNAR1 and IFNAR2 dependent in both human (Fig. 10a, b) and mouse cells (Supplementary Fig. 12). Furthermore, in IL-1α pretreated keratinocytes STAT1 and STAT2 were activated more earlier in response to IFN-α/β (Fig. 10c and Supplementary Fig. 13).

Since IL-36β induced *IFNAR* transcription via IRF1 in human cells (Fig. 6b, e, f), we further analyzed IRF1 expression and its role in IFNAR regulation in response to IL-1. Similar to our findings for IL-36β, IL-1 also increased production of IRF1

mRNA and protein (Supplementary Fig. 10). Additionally, CRISPR/Cas9 gene editing of *IRF1* resulted in diminished levels of both IFNAR1 and IFNAR2 mRNA and protein following IL-1 treatment of human keratinocytes (Supplementary Fig. 14). In conclusion, IL-1 has the capacity to engage the same pathways as those regulated by IL-36 to promote an antiviral state in keratinocytes.

**Discussion**

Infection occurs when a pathogen avoids detection by the host immune system. A well-known mechanism involves the frequent mutations of cold and flu viral proteins that elude detection by previously generated adaptive host immune responses such as virus protein specific antibodies and T cells. Many pathogens take more active measures to inhibit host immunity, including the IFNs. The HSV-1 genome encodes several proteins, e.g., ICP0, ICP27, VP24, and US3, that block activation of IRF3 and downstream type I IFN expression (reviewed in Zeng[16]). Albeit at a much slower rate, the host co-evolves with microorganisms and may over time gain the ability to counteract microbial immune evasion mechanisms. Previously, we hypothesized that the IL-36 cytokines have evolved for this purpose[2]. The here identified pathway involving IL-36 induced expression of IFNAR (Fig. 5 and Supplementary Fig. 4) represents a mechanism whereby IL-36 may achieve this. By promoting higher IFNAR levels in IL-36 activated cells, the cells become more sensitive to IFN (Fig. 9), which in turn may overcome the lower levels of IFN produced by cells infected with virus encoding IRF3 suppressing mechanisms. Thus, in the present model system, while HSV-1 actively diminish the amount of type I IFN produced by infected keratinocytes[16], the cells mount the same immune response due to their higher levels of IFNAR induced by IL-36. Active antagonism of IFN production is common among viruses, many of which replicate in epithelial cells[26]. Our identified pathway involving IL-36 enhancement of IFN responses represent a first line of defense to counteract these immune evasion strategies.

The IL-36 cytokines are conserved in mammalian species suggesting important common functions preserved throughout this part of the animal kingdom[1,2]. The here described IL-36β → IFNAR → STAT1/STAT2 → ISG pathway is present in both human and mouse keratinocytes and may represent one such function. Serendipitously, we observed that HSV-1 infection progressed faster in human keratinocytes (Fig. 4a) than in the mouse cells (Fig. 4b). This likely reflects that HSV-1 has evolved as a human pathogen. Hence, the here used clinical NS isolate has adapted to specifically enter human cells, evade immune responses engaged by the human forms of innate sensors, and utilize the human transcription and protein synthesis machinery for its replication. As mouse protein sequences differ from those in humans, the virus is probably less well adapted for successful infection in the mouse cells. Further studies may reveal the specific proteins responsible for the observed species-specific difference in HSV-1 propagation and identify additional human and mouse pathogens restricted by the here identified pathway.

While the ligand binding IL-1RL2 and IL-1R1 are specific to their respective cytokine, IL-36 and IL-1, they utilize the same co-factor to active the same intracellular signaling cascade[2]. Thus, not surprisingly, we find that IL-1 can activate the same antiviral pathway (Fig. 10 and Supplementary Figs. 10–14) as that here identified engaged by IL-36β. By similarity, we anticipate that IL-36α and IL-36γ also stimulate this mechanism. This raises the intriguing question 'why are there so many cytokines that activate the same pathways?'. A complete answer to this likely encompasses multiple aspects of cytokine biology and tissue physiology. Previously, we proposed that the IL-36 cytokines evolved through

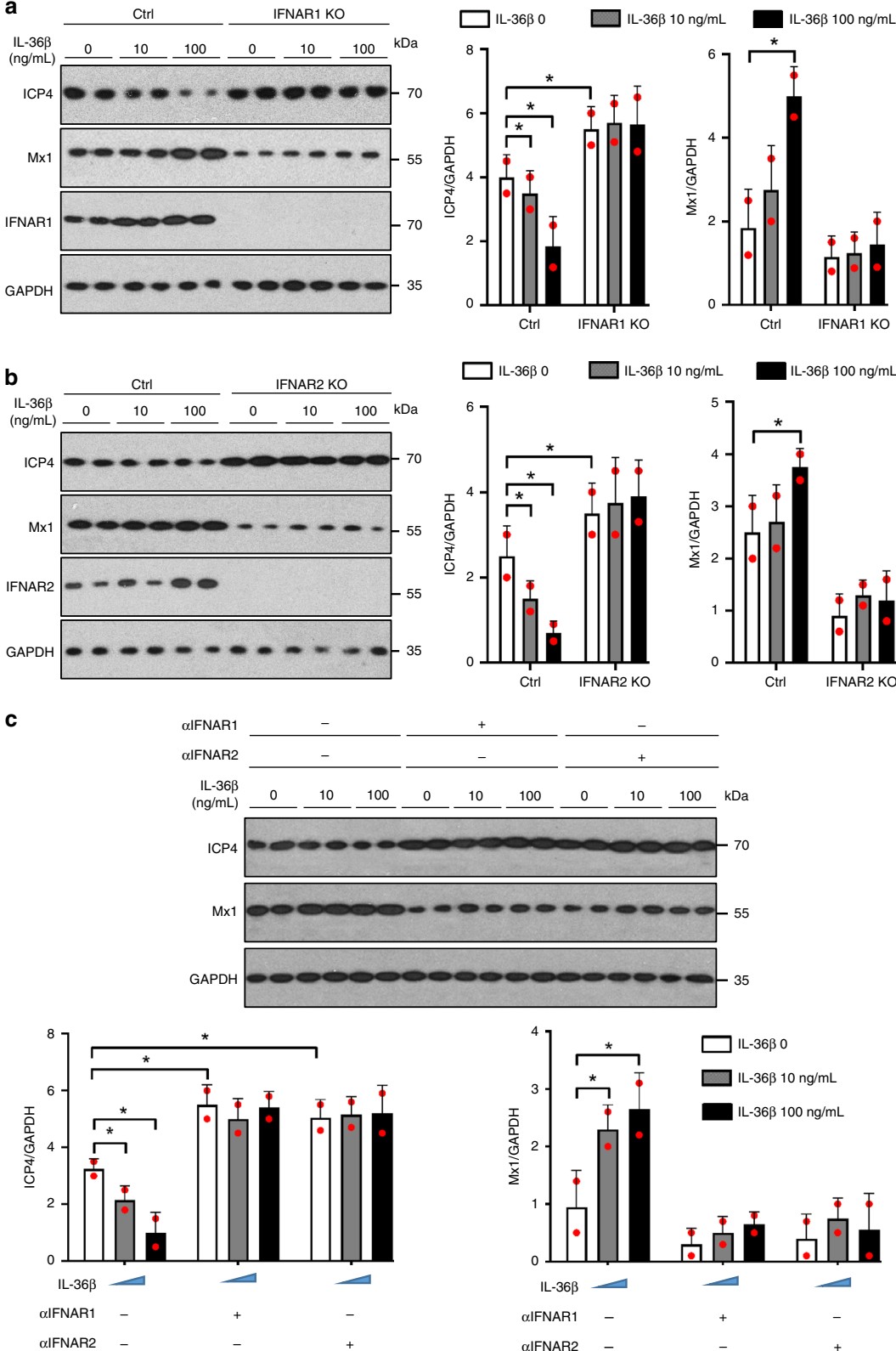

**Fig. 8** IL-36β induced antiviral state is dependent upon IFNAR. **a** Human keratinocytes were transfected with control (Ctrl) or IFNAR1 gRNA/Cas9 expression plasmids, treated with IL-36β and infected with HSV-1. **b** Control (Ctrl) or IFNAR2 gRNA/Cas9 expression plasmid transfected human keratinocytes were treated with IL-36β and infected with HSV-1. **a, b** Levels of ICP4, Mx1, IFNAR1, IFNAR2, and GAPDH were determined using western blotting and ImageJ analyses. **c** Mouse primary keratinocytes were treated with IL-36β as indicated, incubated with neutralizing antibodies against IFNAR or isotype matched Ig and infected with HSV-1. Levels of ICP4, Mx1 and GAPDH were determined using western blotting and ImageJ analyses. **a–c** Quantitative data are shown as means ± SD. *$p < 0.05$ (one-way ANOVA, $n = 2$ biologically independent samples per treatment group). Each red dot represents a single data point. Source data are provided as a Source Data file

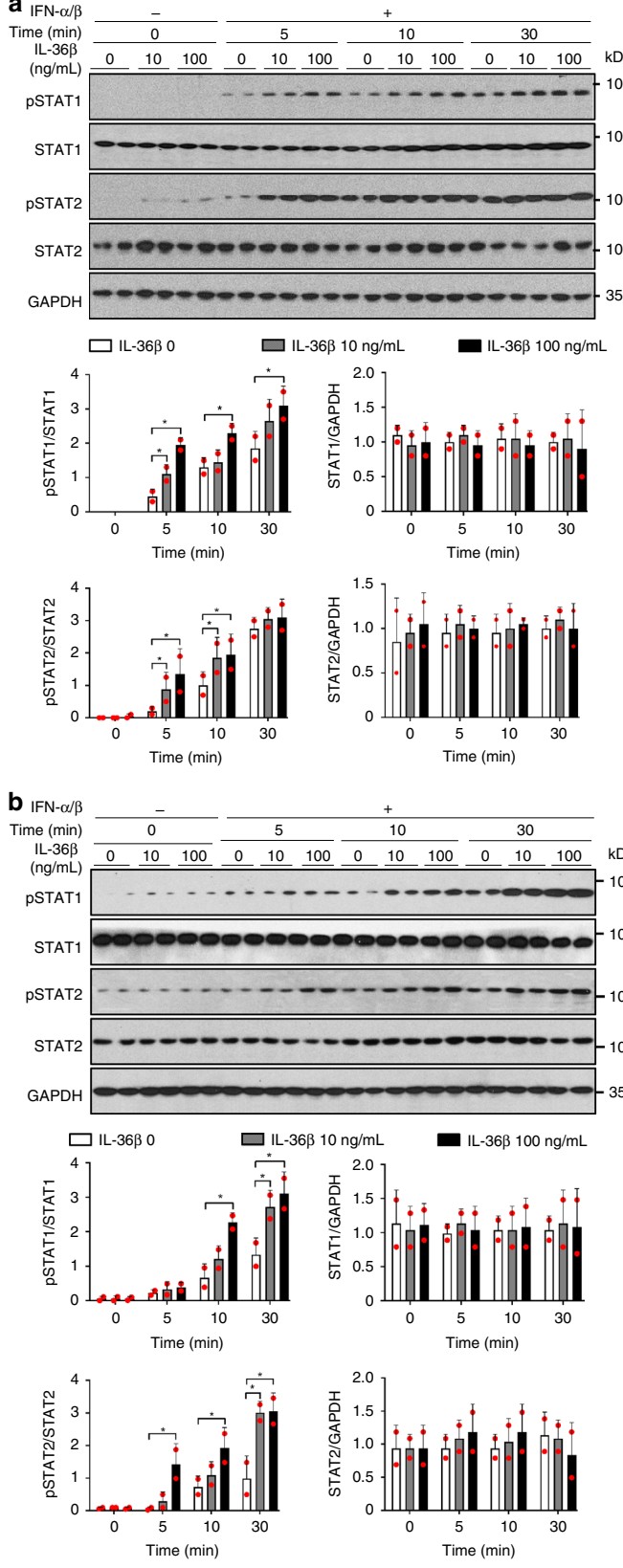

**Fig. 9** IL-36β accelerates type I IFN signaling. Medium only or IL-36β treated keratinocytes were further treated with type I IFN (0.01 ng mL$^{-1}$) as indicated. Phosphorylation of STAT1 and STAT2 was examined by western blotting and ImageJ analyses. Quantitative data are shown as means ± SD. *$p < 0.05$ (one-way ANOVA, $n = 2$ biologically independent samples per treatment group). **a** Human keratinocytes were analyzed. **b** Mouse primary keratinocytes were examined. Each red dot represents a single data point. Source data are provided as a Source Data file

microbial immune evasion. Furthermore, even though the IL-1 family members engage the same pathways at the single cell level, at the tissue and whole organism stage individual cytokines may have adapted to fulfill organ-specific functions. Some of these may be aimed at controlling pathogens targeting certain tissues and other functions may have been acquired to maintain homeostasis in response to other types of stress. This adaptation may, for example, involve the degree to which they are expressed, their activation through protease cleavage and cell types expressing the cytokines and their respective receptors. Future studies of other organ-pathogen systems may reveal such functions and mechanisms.

A recent study, published during the preparation of this manuscript, characterized an IL-1 induced pathway in fibroblasts and endothelial cells that requires IRF1 for induction of ISGs independently of IFNAR[27]. This is in sharp contrast to the mechanism identified in the present study, where IL-36 enhances innate antiviral immunity and ISG expression through IFNAR in keratinocytes (Fig. 8). Having distinct means to overcome immune evasion mechanisms in different cells types may have evolutionary advantages as it can help provide safeguards against the ever-evolving pathogens, i.e., as a microorganism gains a new evasion strategy allowing it to survive longer in one tissue/cell type, alternative host countermeasures in another tissue will contribute to restricting the pathogen. Interestingly, the IL-1 → IRF1 → ISGs pathway in human fibroblasts and endothelial cells is absent in mouse cells[27]. This may represent a gain-of-function in man or a loss-of-function in mice as the two species have adapted to different pathogens.

Having cell type specific approaches for promoting immune responses may not only serve to limit viral spread, but could also contribute to constraining inflammation that when dysregulated may cause chronic diseases such as systemic lupus erythematosus (SLE) and psoriasis. SLE is well-known to be associated with increased expression of ISGs, the IFN signature[28]. A major focus in the SLE field has been on the role of dendritic cells in driving this IFN signature;[28] however, recent work has come to the recognition of keratinocytes as important players in cutaneous lupus pathogenesis[29,30]. ISGs such as Mx1, OAS2, and ISG15 are also upregulated in psoriatic keratinocytes[31] and both conditions are associated with increased IL-36 expression[2,32] and IRF1 activity[33]. Despite these similarities, the skin pathologies are distinct; an improved understanding of cell type specific contributions may shed light on different disease mechanisms and underlying genetic predisposition.

One of the best-established functions of IL-36 is promoting neutrophil recruitment to the skin through induction of chemokines in keratinocytes[34,35]. While it is well-known that HSV lesions in epithelial tissues are neutrophil rich, the function of these neutrophils remains unknown. Previous studies reported that depletion of neutrophils does not affect viral replication or cutaneous lesion development[36], and more recent work found no effect upon T cell expansion and recruitment into the skin[37]. Interestingly, in a mouse model of vaginal HSV-2 infection prophylactic IL-36γ treatment promotes neutrophil recruitment

gene duplication of *IL1B* to counteract microbial immune evasion strategies that prevent activation of IL-1β by utilizing alternative mechanisms of activation and extracellular release[2]. The here identified enhancement of type I IFN (Figs. 9 and 10c) by IL-36 and IL-1 represents a complementary approach to overcoming

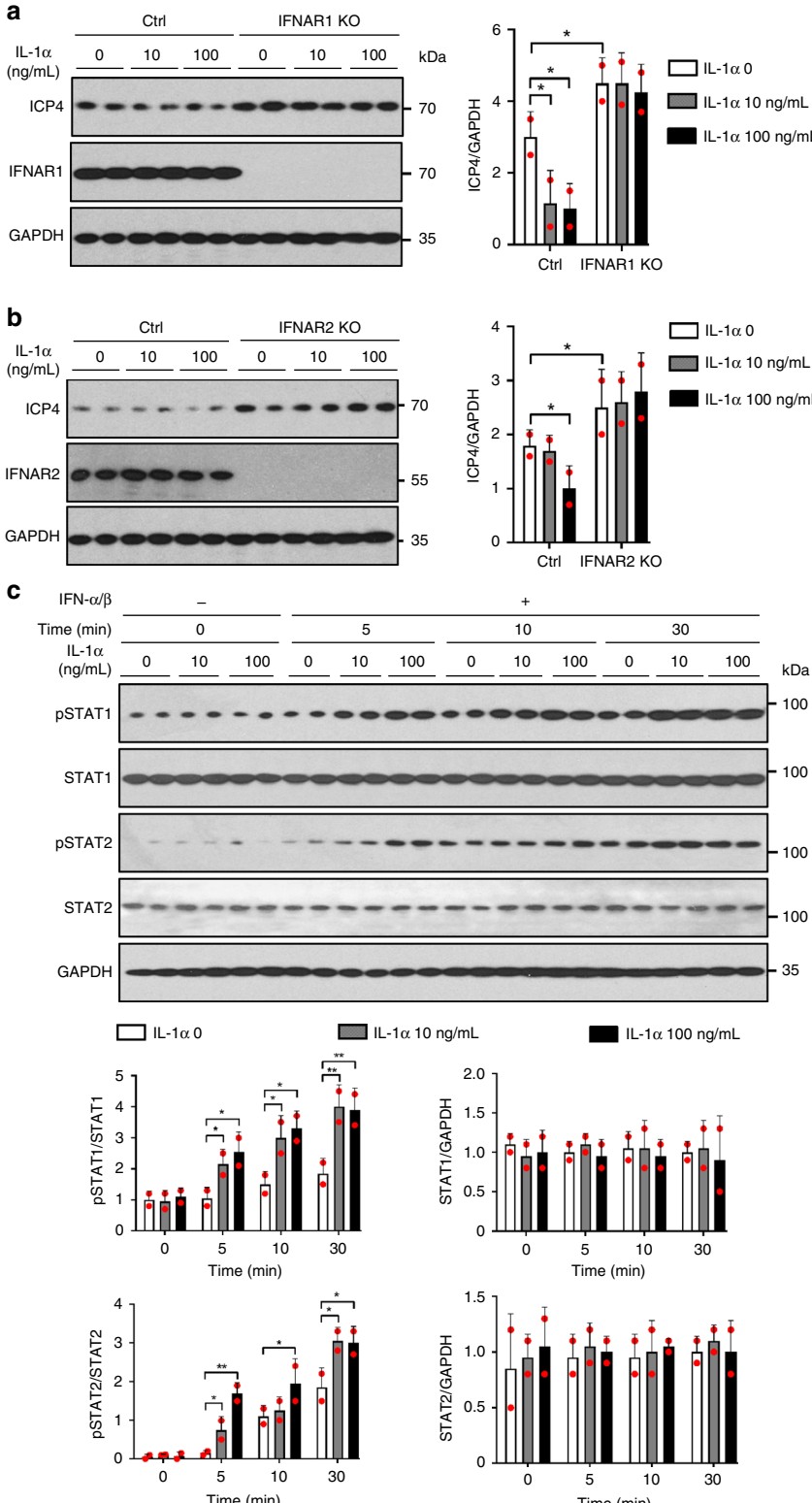

**Fig. 10** IL-1 promotes enhanced type I IFN signaling through IFNAR in human cells. **a**, **b** Gene editing was performed in human keratinocytes using control (Ctrl), *IFNAR1* (**a**) or *IFNAR2* (**b**) gRNAs and Cas9. Cells were treated with medium only or IL-36β, infected with HSV-1 and analyzed by western blotting. **c** Cells were pre-treated with medium only or IL-1α followed by IFN-α/β. STAT1/2 activation was examined by western blotting. Quantitative data are shown as means ± SD. *$p < 0.05$; **$p < 0.01$ (one-way ANOVA, $n = 2$ biologically independent samples per treatment group). Each red dot represents a single data point. Source data are provided as a Source Data file

and protects against pathogenesis[18]. Hence, further studies of the role of IL-36 recruited neutrophils in vaginal mucosa may spread new light on the enigmatic function of these prominent cells in active epithelial HSV infections. It is also noteworthy that mucosal tissues are much more susceptible to HSV infection than skin; therefore, identification of which endogenous IL-1 superfamily members, including the three IL-36s, are functionally active in the vaginal model may improve our understanding of how the cytokine family has evolved to not only act as countermeasures to pathogen immune evasion mechanisms but also to fulfill tissue specific functions.

Previously we observed delayed recruitment of inflammatory cells to emerging zoster-like HSV-1 infection sites in mice deficient in IL-1R1[20]. In the here examined total skin area, levels of CD86 were higher in IL-36β KO mice than wild-type mice (Fig. 1d and Supplementary Table 1). This could reflect delayed recruitment of CD86 expressing antigen presenting cells to the infected skin as these cells must migrate further to lymph nodes to fulfill their functions. We did not pursue this observation further as our previous studies reveal no effect of IL-36β deficiency upon initiation of adaptive immune responses in the present model[17]. It is plausible that the overlapping functions of the IL-36 and IL-1 cytokines provide sufficient redundancy to ensure fully functional outcomes even in the absence of one of these cytokines. However, enhanced recruitment of antigen presenting and T cells may be an additional mechanism whereby prophylactic IL-36γ protects against subsequent HSV-2 challenge[18].

Mechanical injury of the skin is initially required for HSV-1 infection of keratinocytes and neurons in our mouse model. The neutrophil and T cell involving inflammation caused by this tissue damage may contribute to IFN production (Supplementary Figs. 1 and 9) and other immune responses that limit progression of the primary infection. This could explain why we were not able to detect differences in viral load in wild type and IL-36β$^{-/-}$ primary sites 3 days post-infection[17]. In contrast, here we detect more virus 6 days post-infection along the IL-36β KO dermatomes where tissue damage is only caused by the virus and the resulting immune responses (Fig. 1a, b). As the virus reemerges from the neurons and starts to proliferate in the keratinocytes, inflammation is absent (Fig. 3). In these local microenvironments, low levels of type I IFN produced by the keratinocytes may be critical for restricting the spread of the virus to neighboring cells. This would be in agreement with the observed pSTAT1 staining in keratinocytes at or near early and intermediate HSV-1 lesions in our model (Fig. 3a–p). Future development of methods to monitor progression of viral infection concurrent with immune activation may allow distinction between cells that survive and die. Such approaches may also contribute insight into reactivation of HSV from latency.

In summary, we have identified a novel mechanism whereby IL-36 promotes an antiviral state in epithelial cells by upregulating the expression of the type I IFN receptor complex and increasing sensitivity to IFN. Such a mechanism may have evolved to control viruses that inhibit innate immunity by blocking production of type I IFN.

## Methods

**Cytokines and antibodies**. Recombinant human IL-36β (Catalog #: 6834-ILB-025), mouse IL-36β (Catalog #: 7060-ML-010) protein, and Universal Type I IFN (Catalog #: 11200–1) were obtained from R&D Systems. Recombinant human IL-1α (Catalog #: 200–01 A) and mouse IL-1α (Catalog #: 211–11 A) were from PeproTech. Mouse monoclonal antibodies to HSV-1 ICP4 (H943, Catalog # sc-69809, RRID:AB_844234, used at 1:500 dilution), IFN-α/βRα Antibody (H-11, Catalog # sc-7391, RRID:AB_2122749, used at 1:800 dilution), IFN-α/βRβ Antibody (F-7, Catalog # sc-137209, RRID:AB_2122750, used at 1:500 dilution) and Mx1 Antibody (E-8, Catalog # sc-398564, RRID:AB_1146318, used at 1:600

dilution) were acquired from Santa Cruz Biotechnology. Rabbit monoclonal antibodies to phospho-STAT1 (Tyr701, 58D6, Catalog # 9167, RRID: AB_561284, used at 1:800 dilution (Westerns) and 1:50 dilution (immunohistochemistry)), phospho-STAT2 (Tyr690, D3P2P, Catalog # 88410, RRID: AB_2800123, used at 1:800 dilution), STAT1 (D1K9Y, Catalog # 14994, RRID:AB_2799965, used at 1:1000 dilution (Westerns)), STAT2 (D9J7L, Catalog # 72604, RRID:AB_2799824, used at 1:1000 dilution), Gapdh (14C10, Catalog # 2118, RRID: AB_561053, used at 1:2000 dilution), IRF1 (D5E4, Catalog # 8478, RRID:AB_10949108, used at 1:1000 dilution), anti-rabbit IgG HRP-linked (Catalog # 7074, RRID: AB_2099233) or anti-mouse IgG HRP-linked (Catalog # 7076, RRID: AB_330924) antibodies (used at 1:10,000 dilution) were obtained from Cell Signaling Technology.

**Mice**. All procedures involving mice were approved by the Temple University Institutional Animal Care and Use Committee and in compliance with the U.S. Department of Health and Human Services Guide for the Care and Use of Laboratory Animals. Mice were housed in a specific pathogen free facility. The IL-36β$^{-/-}$ mouse strain was characterized previously[17,35] and is currently maintained on the C57BL/6 J background. For genotyping, 2 mm ear punches were boiled in 10 mM NaOH, 0.1 mM EDTA for 10 min, neutralized with 1 M Tris, pH 8.0 and used for PCR with primers: gIL1F8(SU) CTTAGGGATTGCTGTCCTTG, gIL1F8 (LacinZRev) GTCTGTCCTAGCTTCCTCACTG, gIL1F8(TUR) ATGCCACC-TACCAGGCTTGAC[38]. Wild type C57BL/6 J, Irf1$^{-/-}$ and Ifnar1$^{-/-}$ mice were obtained from the Jackson Laboratory, Bar Harbor, ME, and bred in-house. The Stat1$^{-/-}$ and Stat2$^{-/-}$ mice, both on the C57BL/6 background, were provided by Dr. Ana Gamero at Temple University Lewis Katz School of Medicine. Stat1$^{-/-}$ were generated by Dr. David Levy[39] and Stat2$^{-/-}$ mice were generated by Dr. Christian Schindler on the SvJ background[40] that Dr. Gamero backcrossed 10 generations onto the C57BL/6 genetic background[41].

**In vivo HSV-1 infections**. The original stock of the clinical HSV-1 isolate NS was obtained from Dr. Harvey Friedman (University of Pennsylvania, Philadelphia, PA). The virus was expanded in Vero cells (ATCC) infected at a low MOI in serum free DMEM. Infected cultures were collected 48 h post-infection. Mice were used at 8 weeks of age and matched for sex in each individual experiment. The number of mice per group in each experiment is provided in the figure legends. Mice were denuded the day before infections by sequential fur trimming and epilating cream application. Cream was gently removed with water. Scratch inoculations were performed with $1.5 \times 10^6$ PFU HSV-1 on the right flank. Mice were photographed next to a ruler on indicated days and lesions sized using the ImageJ software https://imagej.nih.gov/ij/ and the pictured ruler for scaling. Mice were evaluated for paralysis using splaying reaction in hind legs and paws upon lifting by the tail.

**Viral DNA copy numbers**. Infected skin regions of equal sizes were collected, measured and homogenized. DNA was extracted using the Qiagen DNeasy Blood & Tissue kit according to manufacturer's instructions (Qiagen). HSV-1 genome copy numbers were determined using quantitative real-time PCR with primers HSVgD-F CTACTATGACAGCTTCAGCGA and HSVgD-R CCGTCCAGTCG TTTATCTTC, the TaqMan probe VIC-CAGTTATCCTTAAGGTCTC-MGNFQ and TaqMan Gene Expression Master Mix (Applied Biosystems). A previously characterized plasmid[20] containing the gD PCR fragment was used for generating a standard curve.

**RNA isolation, PCR arrays, and real-time PCR**. RNA was isolated using RNeasy Plus (Qiagen) and reverse transcribed using random hexamers as primers and AMV reverse transcriptase (Promega) according to the manufacturer's instructions[42]. The RT2 Profiler PCR Array Mouse Antiviral Response (PAMM-122Z, Qiagen) was used according to the manufacturer's instructions and analyzed using the Data Analysis Center (Qiagen) resources. For real-time PCR analyses primers listed in Supplementary Table 2 were used.

**Histology and immunohistochemistry**. Skin was fixed in formaldehyde and processed for histology and immunohistochemistry at the Histopathology facility at Fox Chase Cancer Center in Philadelphia. Antigen retrieval was performed using EDTA and slides were stained for pSTAT1 using rabbit monoclonal phospho-STAT1 (Tyr701) (58D6).

**Cell cultures**. The Vero (ATCC) and HaCaT (obtained from Dr. Meenhard Herlyn, Wistar Institute, Philadelphia, PA) cell lines were maintained using standard procedures in DMEM supplemented with 10% fetal bovine serum and Gentamicin. Primary mouse keratinocytes were isolated from 24 to 72 h old pups[20]. The pup tail tips were boiled in 10 mM NaOH, 0.1 mM EDTA, neutralized with 1 M Tris, pH 8.0 and used for PCR typing of sex with primers SMCX-1, 5′-CCGCTGCCAAATTCTTTGG-3′ and SMC4-1 5′-TGAAGCTTTTGGCTTTG AG-3′. These primers generate 1 product from female cells and 2 products from male cells. Skins were floated in 0.25% Trypsin (Invitrogen) overnight at 4 °C. The epidermis and dermis were separated and keratinocytes from the epidermis dissociated by mechanical shearing. Keratinocytes were maintained in a 1:1 mix of Keratinocyte-SFM with calcium and Keratinocyte-SFM without calcium

(Invitrogen) supplemented with 10 ng mL$^{-1}$ mouse epidermal growth factor (Sigma-Aldrich), 200 µg mL$^{-1}$ bovine pituitary extract (Sigma-Aldrich) and Gentamicin. Experiments were performed with independent duplicate samples per timepoint and treatment. These duplicate samples were analyzed separately.

**Gene editing**. HaCaT cells were grown to 70–80% confluence, detached with Trypsin (0.25%, Invitrogen), washed with OPTI-MEM (Invitrogen) and transfected using electroporation[42] with transEDIT CRISPR All-in-one gRNA expression vectors (Transomic) targeting IFNAR1 (Tevh-1135347-Pclip-ALL-Hcmv-puro), IFNAR2 (Tevh-1086376-Pclip-ALL-Hcmv-puro) or IRF1 (Tevh-1123055-Pclip-ALL-Hcmv-puro). Control cells received the scrambled sequence construct tela1015-CRISPR-NT-#1-Pclip-ALL-Hcmv-puro. Transfections were performed in 4 mm electroporation cuvettes using two 8 msec pulses of 200 V at a 100 msec interval. Targeted cells were selected with puromycin (2 ng mL$^{-1}$) for 4–5 days before experimental analyses.

**Western blotting and ImageJ quantification**. The cells were lysed in 50 mM HEPES, 150 mM NaCl, 20 mM β-glycerophosphate, 1 mM NaVO$_4$, 1% NP-40, 1 mM Benzamidine, 1 mM EDTA, 50 mM NaF, 20 mM DTT, mammalian protease inhibitor (Sigma), pH 7.5. Electrophoresis was in 12% denaturing polyacrylamide gels that were transferred to polyvinylidene fluoride (PVDF) and reacted with appropriate antibodies. The protein bands were detected with secondary antibodies conjugated to HRP and enhanced chemiluminescence. Films were scanned, and band intensities measured using ImageJ. GAPDH expression was used as a loading control. Uncropped and unprocessed scans are included in the Source Data file.

**Statistical analyses**. All experiments were performed at least three times (independent) unless otherwise indicated. All data shown are arithmetic means ± standard deviations unless indicated otherwise. Statistical significance was evaluated using One-Way ANOVA with post hoc tests unless started otherwise.

**Reporting summary**. Further information on research design is available in the Nature Research Reporting Summary linked to this article.

## Data availability

All data generated and analyzed during this study are included in this published Article and its Supplementary Information and Source Data files. The source data underlying Figs. 1a–c, f, 3–10, and Supplementary Table 1, Supplementary Figs. 2–14 are provided as a Source Data file.

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

## Acknowledgements

This research was made possible through funding from The National Institute of Allergy and Infectious Diseases (R01 grant AI125111 to LEJ) and Temple University Bridge Funds (AMG). The Histopathology facility where tissues were processed for H&E staining and immunohistochemistry is supported by The National Cancer Institute core grant P30 CA006927. We would like to thank Dr. Siva Uppalapati for assistance with in vivo studies.

## Author contributions

Conceptualization by L.E.J. Investigation by P.W.; Writing – Original Draft by P.W. and L.E.J.; Writing – Review & Editing by A.M.G. and L.E.J.; Figures and Source Data by P.W.; Funding acquisition by L.E.J.; Resources by A.M.G. and L.E.J.; and Supervision by L.E.J.

## Additional information

**Competing interests:** The authors declare no competing interests.

