## [Peer Review File · Nature Communications]

Reviewers' comments:

Reviewer #1 (Remarks to the Author):

The authors have examined the role of IL-36 β in herpes simplex virus (HSV) skin infection. They previously showed that IL-36 β deficient mice have more severe skin lesions. In the present manuscript they show that IL-36 β deficient mice have increased viral load with decreased levels of class I IFN signature genes. They showed that IL-36 β regulates IFNAR1 and IFNAR2 mRNA and protein levels as well as IRF1 levels in cultured human and mouse keratinocytes. IRF1 deficiency has only partial effect on IL-36 β -induced IFNAR expression in mouse keratinocytes. IL-36 β increases the effect of IFN α/β on STAT1/STAT2 activation in keratinocytes.

The manuscript is clearly written and provides original results on the role of IL-36 in epithelial host defense mechanisms against viral infections via enhancing class I IFN responses. The effect of IL-36 signaling seems mediated by an increased expression of IFN receptors in keratinocytes. The other strength of the manuscript is the inclusion of studies using both mouse and human cells.

The authors concentrated their studies on IL-36 β , which is a logical choice according to their previous results. However, as IL-1 and IL-36 share common signaling pathways it would have been interesting to test in parallel the effect of IL-1 using the same readouts in human and mouse keratinocytes.

In figure 2E the authors show IHC on pSTAT in mouse skin following HSV infection. The positive staining in neutrophils is easily seen. However, the presence of pSTAT in keratinocytes is not clearly demonstrated. Furthermore, the results obtained in IL-36 β KO mice should be included to see if there is a difference of staining. A negative control using STAT KO mice would be also useful to assess the specificity of the immunostaining.

The effect of IRF1 deficiency on survival and skin lesion severity is puzzling. From the data in mouse keratinocytes the effect of IL-36 signaling on IFNAR expression seems largely IRF1 independent. Furthermore, the skin viral load is not significantly influenced by IRF1. Thus, the role of IRF1 might be independent of the clearance of HSV. The authors propose that IRF1 KO mice may have increased systemic HSV infection but do not provide any evidence of this? Did they measure viral load in blood or other organs? What about exaggerated inflammatory responses in IRF1 deficient mice independent of viral clearance?

Reviewer #2 (Remarks to the Author):

In this paper the Jensen lab follow up their previous studies of the mechanism of an antiviral effect of interleukin 36 β against herpes simplex virus infection in the mouse zosteriform model. Previously they showed that HSV1 induced IL36 α and β in mouse skin (with IL36 γ unchanged). Knockout of IL36 β enhanced HSV lesion size and reduced survival in this model without affecting adaptive immunity. Here they show evidence that IL36 β increases infected keratinocyte sensitivity to type I interferons via increased interferon receptor (IFNAR1 and 2) expression by signaling through STAT1 and 2.

The significance of the work is assignment of a key function to IL36 β in antagonizing viral immune-evasive mechanisms which inhibit interferon action. A strength of the work is the follow up of in vivo mouse models with direct studies of the key HSV target cells, keratinocytes, of both mouse and human origin.

In general the studies are well done and the data mostly support the above pathway for anti-HSV effects of IL36 β .

In figure 1 the effect of IL36 β was examined on skin virus DNA and ICP4 at day 6. Was there any

effect on replication in the dorsal root ganglion? Or on the primary skin lesions at day 2? Early effects of interferon are most important.

In Figure 3B were they surprised that 'early antigen' ICP4 did not appear until after 12 hpi in mouse keratinocytes, unlike in human cells. Is there any reason for the discrepancy?

In figure 5 the statement in Results that IRF1 plays a role in upregulation of IFNAR1 in mouse and human keratinocytes is somewhat confusing and not really supported by the figure. In figure 5D there is no significant difference in the effects of IL36 β on keratinocyte expression of IFNAR 1 and 2 proteins between control or IRF1 KO mice. ie IRF1 only plays an unambiguous and significant role in IL36 β stimulation in IFNAR1 and 2 upregulation in human keratinocytes as shown in Figure 5F. This interpretation is also more consistent with results in Figure 6. The reduction in IFNAR1 and 2 in human IL36 β KO constitutively also does not relate to HSV effects.

In figure 8 how do the very low levels of type I interferon used match with those released from HSV infected keratinocytes or Langerhans cells in the epidermis?

Discussion:

The report from Gardner et al showing that IL36 γ inhibits HSV replication after vaginal application in mice and enhances survival (and also enhances proinflammatory cytokines and polymorph infiltration as they mention) appears to contradict their findings with IL36 γ in their previous report (2017). Can they explain this? Have they tested for individual or combinatorial antiviral effects of other IL36 isoforms in their keratinocyte systems?

Reviewer #3 (Remarks to the Author):

The manuscript authored by Peng Wang, Ana M. Gamero and Liselotte E. Jensen investigates the role of IL-36-beta in the innate antiviral response to cutaneous herpes simplex virus type 1 (HSV-1) using a model involving the scarification and replication of HSV-1 in the flank of mice. The manuscript reports a very carefully performed and evaluated study of the mechanisms associated with IL-36-beta expression on the type 1 interferon (IFN) response both in the skin of IL-36-beta knockout (KO) and wild-type mice, and in isolated mouse and human keratinocytes in vitro.

The study presents compelling evidence that IL-36-beta induces the expression, at both the mRNA and protein levels, of the two chains of the type I IFN receptors in mouse and human keratinocytes, and that Interferon Response Factor 1 (IRF1) is also upregulated by IL-36-beta, which in turn upregulates type I IFN receptor expression, although IRF1-dependent antiviral mechanisms are expressed in human keratinocytes, but that the mechanisms are IRF1-independent in mouse keratinocytes.

Overall, the manuscript provides evidence of a clear link between IL-36-beta expression and the optimization of innate IFN-mediated antiviral responses to cutaneous HSV-1 infections. The manuscript addresses the role of IL-36-beta in overcoming the immune evasion strategies of HSV-1 to limit IFN-mediated antiviral responses, and links this to the more accelerated and higher levels of expression of the phosphorylated (activated) forms STAT1 and STAT2 transcription factors. The study provides important evidence that explains the role of IL-36-beta in this process.

RESPONSE TO REVIEW OF MANUSCRIPT NCOMMS-19-01666

We would like to thank the reviewers for their insightful comments, which have helped clarify experiments, conclusions and significance. Several new experiments have been performed resulting in 9 completely new figures (Figure 3, Figure 10, Supplementary Figure 1 and Supplementary Figures 9-14), and two new panels in Figure 7. Point-by-point responses to review comments are listed below and changes are highlighted in yellow in the text.

Editors summary

Referee 1's comments:

- 1) relating to the need to test the effect of IL-1 in the human and mouse keratinocytes in order to compare the overlap with IL-36 signalling pathways.

RESPONSE: To address this, we have performed several new series of experiments with IL-1 and both human and mouse cells. The results are shown in new Figure 10 and new Supplementary Figures 10-14, and described in the results section 'IL-1 also enhances IFN- α/β signaling via IFNAR upregulation'. Further details are described below in direct response to reviewer 1.

- 2) Additionally, please address this referee's comments about the need to demonstrate the presence of pSTAT in keratinocytes, and analysis of knockout mice.

RESPONSE: This is addressed in detail below in response to the reviewer. Briefly, new *in vivo* experiments have been performed and Figure 3 and Supplementary Figure 1 have been added to illustrate the outcomes. The experiments are described in section 'Results: IL-36 β enhances STAT1/2 activation in keratinocytes'.

- 3) Finally, please address this referee's comments asking for experimental evidence to support your conclusions that IRF1 KO mice may have increased systemic HSV infection.

RESPONSE: We have changed the conclusion from 'IRF1 is required for systemic immunity and/or preventing dissemination (Figure 7A)' to 'IRF1 plays a critical role in determining disease outcome (Figure 7A-D)'. We have also added all the both existing and new data we have to support the notion that the mice are dying from systemic disease in new figure panels Figure 7B-C. A more detailed response is provided below to the reviewer.

Referee 2's concerns about

- 1) the data in Figure 5 not supporting the conclusions that IRF1 plays a role in upregulation of IFNAR1 in mouse and human keratinocytes.

RESPONSE: The text has been rewritten to better reflect our intended interpretation. See also response to the reviewer below.

- 2) Please also address this referee's comments and requests for clarification about a number of the figures in your manuscript.

RESPONSE: Please see below how these have been addressed.

Reviewer 1

Comment: The authors concentrated their studies on IL-36 β , which is a logical choice according to their previous results. However, as IL-1 and IL-36 share common signaling pathways it would have been interesting to test in parallel the effect of IL-1 using the same readouts in human and mouse keratinocytes.

RESPONSE: As mentioned above in response to the editor several new series of experiments have been performed using both human and mouse cells. We focused specifically on determining whether IL-1 would activate the same pathways as IL-36. The experiments are described in the new 'Results' section 'IL-1 also enhances IFN- α/β signaling via IFNAR upregulation' and data are shown in Figure 10 and Supplementary Figures 10-14. Briefly, we find that IL-1 does activate the same mechanisms. Since this raises interesting questions relating to why the IL-1/IL-36 system is so apparently redundant the data is also discussed in the third paragraph of the 'Discussion' in a context that also aims to address comment from reviewer 2 regarding IL-36 γ in the vaginal mouse model of HSV infection.

Comment: In figure 2E the authors show IHC on pSTAT in mouse skin following HSV infection. The positive staining in neutrophils is easily seen. However, the presence of pSTAT in keratinocytes is not clearly demonstrated. Furthermore, the results obtained in IL-36 β KO mice should be included to see if there is a difference of staining. A negative control using STAT KO mice would be also useful to assess the specificity of the immunostaining.

RESPONSE: We have performed several new *in vivo* infections to address this. We used wild type and STAT1 KO mice to demonstrate the specificity of the antibody, and staining in keratinocytes. Since the STAT1 KO mice are moribund on day 4 post-infection, we used primary infection sites for this. This data is shown in new Supplementary figure 1 and described in the

first paragraph of 'Results: IL-36 β enhances STAT1/2 activation in infected keratinocytes'. We are glad we did this as it revealed an issue with non-specific staining around neutrophils. We have not been able to establish a dual-staining protocol for detecting pSTAT1 and keratinocyte markers at the same time; hence, we have taken the next best approach and shown consecutive tissue sections with H&E staining and pSTAT1 to help better identify the pSTAT1 positive keratinocytes.

We also performed new experiments with wild type and IL-36 β KO mice. Since we observe differences in HSV-1 DNA and ICP4 in skin along the dermatome, we focused on these sites. The approach and data are described in the second paragraph of 'Results: IL-36 β enhances STAT1/2 activation in infected keratinocytes' and shown in the new Figure 3. The H&E approach helped identify HSV infected cells due to the cytopathic effects caused by the virus. Hence, we now show different stages of infection progression in the epidermis. We furthermore focused specifically on the very early lesions/infection sites where inflammation was not yet present to, as best possible, match the cell culture models. The observation that pSTAT1 is present in proximity of infected cells *in vivo* (Figure 3) correlate with our use of MOI = 0.01 *in vitro*, i.e. a model where infected cells have time to cause changes in neighboring cells. These microenvironments, where infected cells interact with neighboring cells, are briefly discussed in the second to last paragraph of the discussion. Significant model development will be required for further studies of this.

Comment: The effect of IRF1 deficiency on survival and skin lesion severity is puzzling. From the data in mouse keratinocytes the effect of IL-36 signaling on IFNAR expression seems largely IRF1 independent. Furthermore, the skin viral load is not significantly influenced by IRF1. Thus, the role of IRF1 might be independent of the clearance of HSV. The authors propose that IRF1 KO mice may have increased systemic HSV infection but do not provide any evidence of

this? Did they measure viral load in blood or other organs? What about exaggerated inflammatory responses in IRF1 deficient mice independent of viral clearance?

RESPONSE: It is true that IRF1 could be a negative regulator of inflammation, and therefore the absence of IRF1 would lead to more tissue damaging inflammation. Since we have no evidence to support this, we have simply revised the first sentence of the last paragraph in 'Results: The role of IRF1 in antiviral immunity is species dependent' from 'IRF1 is required for systemic immunity and/or preventing dissemination (Figure 7A)' to 'IRF1 plays a critical role in determining disease outcome (Figure 7A-D)'.

Several groups have observed gastrointestinal dysfunction in mouse models of HSV infection. One lab reported that mortality in at least the vaginal model is due to constipation (toxic megacolon; Khoury-Hanold et al., 2016). We have added text in the first paragraph of 'Results: The role of IRF1 in antiviral immunity is species dependent' to explain this. We have also added new data panels to Figure 7, specifically panels B and C showing respectively weight loss and increased frequency of gastrointestinal dysfunction in the IRF1 KO mice at day 9 post-infection. Because the gastrointestinal phenotype in the mice generally, if at all, does not appear in humans, this is not a phenomenon we have the financial or human resources to pursue further. However, we felt obliged to disclose that we did see increased mortality in the mice despite our negative findings regarding a role in innate skin immunity. Due to our limited resources we did not determine viral loads in blood or other organs. Analyses of IRF1s protective function(s) against lethal outcome represent a very different, and separate, type of study than that reported here.

We sincerely hope these changes clarify that we are not making any conclusions to why IRF1 KO mice have increased mortality; our data only support the conclusion that IRF1 is not involved in innate immunity in the skin.

Reviewer 2

Comment: In figure 1 the effect of IL36 β was examined on skin virus DNA and ICP4 at day 6. Was there any effect on replication in the dorsal root ganglion? Or on the primary skin lesions at day 2? Early effects of interferon are most important.

RESPONSE: In our previous published study (Milora et al 2017, Sci. Rep.), we examined the primary skin lesions at day 3 and found no differences between wild type and IL-36 β KO mice. Day 3 was chosen in those studies because we observed large variation in titer values within single groups at day 2; an observation that may be due to remnants of the initial HSV-1 inoculum as the mice are infected with 1.5×10^6 PFUs. The reason we are not able to detect differences in the primary infection sites may be related to another of the review comments, i.e., the levels of IFN produced by the skin upon injury/infection (see also below). In the flank model we have to injure the skin for the virus to enter in the primary site. This results in a lot of inflammation that may contribute to restricting the virus. In contrast, the secondary zosteriform lesions are “pure” HSV infections and this may allow us to detect differences between strains. This is now briefly mentioned in ‘Results: IL-36 β enhances STAT1/2 activation in keratinocytes’ and further discussed in the second to last paragraph of the ‘Discussion’.

While we have identified a collaborator, who can teach us how to collect the dorsal root ganglion, we still need to setup our BSL2 facility with a dissection microscope for collecting these tiny structures safely and accurately. Hence, analyses of HSV infected dorsal root ganglion is not physically possible for us at this time.

Comment: In Figure 3B (now Figure 4B) were they surprised that ‘early antigen’ ICP4 did not appear until after 12 hpi in mouse keratinocytes, unlike in human cells. Is there any reason for the discrepancy?

RESPONSE: This outcome actually makes sense since HSV-1 is a human pathogen and we use a clinical isolate of HSV-1, not a mouse or lab adapted virus. Hence, the virus has evolved to attached, enter, evade immunity and propagate utilizing human proteins. Text has been added in the second paragraph of the discussion to explain this.

Comment: In figure 5 the statement in Results that IRF1 plays a role in upregulation of IFNAR1 in mouse and human keratinocytes is somewhat confusing and not really supported by the figure. In figure 5D there is no significant difference in the effects of IL36 β on keratinocyte expression of IFNAR 1 and 2 proteins between control or IRF1 KO mice. ie IRF1 only plays an unambiguous and significant role in IL36 β stimulation in IFNAR1 and 2 upregulation in human keratinocytes as shown in Figure 5F. This interpretation is also more consistent with results in Figure 6. The reduction in IFNAR1 and 2 in human IL36 β KO constitutively also does not relate to HSV effects.

RESPONSE: It is important to note that at the mRNA level there is a significant effect upon mouse *Ifnar1* mRNA after *Irf1* knockout (previous Fig. 5C, now Fig. 6C); however, this effect is insufficient to affect protein levels. It was our intent in the previous version to reflect this subtlety; however, having read through the text again we see how the confusion may have arisen. We have inserted a 'mRNA' and rewritten the last sentence of the first paragraph of 'Results: IRF1 contributes differentially to IFNAR expression' to 'However, this latter effect appears insufficient to affect functional outcome as protein levels are not impacted by the absence of IRF1'. At the end of the second paragraph, we rewrote the last sentence to now read 'In contrast, IRF1 does not impact mouse IFNAR1 and IFNAR2 protein levels.' We sincerely hope that we have identified the text the reviewer is referring to.

Comment: In figure 8 how do the very low levels of type I interferon used match with those released from HSV infected keratinocytes or Langerhans cells in the epidermis?

RESPONSE: We performed new experiments using both wild type and IFNAR1 KO mice to determine how much IFN α and IFN β is released by mock and HSV-1 infected skin. In the first experiment involving wild type mice, we were not able to detect any IFN β , prompting us to use IFNAR1 KO mice instead. We used both an explant and a whole tissue approach for this. While we could detect some IFN β in explant medium, we were not able to detect it in whole tissue. The former positive data is now included in Supplementary Figure 9. Because the explant approach requires the skin to be injured for collection, we seem to have a fairly high background production of IFN β using this approach (note that STAT1 is NOT activated in uninfected skin (Supplementary Figure 1)). How this may explain why we cannot detect differences in viral titers in primary lesions (Milora et al 2017, Sci. Rep., see also our response to the earlier review comment) but can in skin surrounding zosteriform lesions (Figure 1A-B) is discussed in the second to last paragraph of the 'Discussion'. Since the HSV-1 infected skin secreted approximately 60 pg/mL over a 24-hour period, the levels of type I IFN used in our experiments seem well within physiological relevant concentrations, especially in local microenvironment as those shown in Figure 3. This is also discussed in the second to last paragraph of the 'Discussion' and a sentence has been added to the 'Results: IL-36 β accelerates STAT activation in response to type I IFN'.

We were not able to detect any IFN α , but since we cannot for certain exclude the possibility that this was due to limited sensitivity of the assay (detection limit approx. 10-40 pg/mL) or other technical issues, we have not included this negative data.

Comment: Discussion: The report from Gardner et al showing that IL36 γ inhibits HSV replication

after vaginal application in mice and enhances survival (and also enhances proinflammatory cytokines and polymorph infiltration as they mention) appears to contradict their findings with IL36 γ in their previous report (2017). Can they explain this? Have they tested for individual or combinatorial antiviral effects of other IL36 isoforms in their keratinocyte systems?

RESPONSE: Gardner et al used recombinant IL-36 γ designed to be already active, i.e., with a truncated N-terminus. Our previous published study involved IL-36 γ knockout mice, and endogenous IL-36 γ in the wild type mice would have to be activated by a protease. We have now added data showing that IL-1 activates the same pathways as those already shown to be engaged by IL-36 β , as described in the present study. Due to the use of the same receptor, IL-36 γ would be expected to activate the same pathway as IL-36 β ; this is now stated in the discussion. For that reason, we removed 'β' from the manuscript title. Why specifically IL-36 β is so critical in our flank skin model is still a mystery we need to solve; however, it may be related to activation mechanisms, incl. expression and/or protease cleavage. Furthermore, there may be tissue specific responses and activation mechanisms. It is noteworthy that skin is typically much more resistant to HSV infection than mucosal sites. These issues are now discussed further in paragraphs 3 and 6 of the discussion.

Reviewer 3

No changes suggested

Additional changes to comply with editorial policies:

- 1) The title has been tweaked a little to fit within length restriction.
- 2) Some subheadings have been shortened to be 60 characters or less.
- 3) The abstract has been slightly changed to accommodate new findings.

REVIEWERS' COMMENTS:

Reviewer #1 (Remarks to the Author):

The authors have addressed appropriately my comments and questions. No further comments.

Reviewer #2 (Remarks to the Author):

The author responses are reasonable and plausible and introduce the uncertainties in their data and conclusions which are yet to be resolved. However in Discussion para 3 page 15 they make the assumption that their clinical strain of HSV is less well adapted for mouse than human keratinocytes which is not proved. More caution is needed so 'probably' ...is less well adapted... should be inserted.

RESPONSE TO REVIEWERS' COMMENTS:

Reviewer #1 (Remarks to the Author):

The authors have addressed appropriately my comments and questions. No further comments.

RESPONSE: No changes made.

Reviewer #2 (Remarks to the Author):

The author responses are reasonable and plausible and introduce the uncertainties in their data and conclusions which are yet to be resolved. However in Discussion para 3 page 15 they make the assumption that their clinical strain of HSV is less well adapted for mouse than human keratinocytes which is not proved. More caution is needed so 'probably' ...is less well adapted... should be inserted.

RESPONSE: The word 'probably' has been inserted.